# Regionalized regulation of actomyosin organization influences cardiomyocyte cell shape changes during chamber curvature formation

Dena M. Leerberg [1], Gabriel B. Avillion [1], Rashmi Priya[2], Didier Y. R. Stainier [2] & Deborah Yelon [1] ✉

Cardiac chambers emerge from a heart tube that balloons and bends to create expanded ventricular and atrial structures, each containing a convex outer curvature (OC) and a recessed inner curvature (IC). The cellular and molecular mechanisms underlying the formation of these characteristic curvatures remain poorly understood. Here, we demonstrate in zebrafish that the initially similar populations of OC and IC ventricular cardiomyocytes diverge in the organization of their actomyosin cytoskeleton and subsequently acquire distinct OC and IC cell shapes. Altering actomyosin dynamics hinders cell shape changes in the OC, and mosaic analyses indicate that actomyosin regulates cardiomyocyte shape in a cell-autonomous manner. Additionally, both biomechanical cues and the transcription factor Tbx5a influence the basal enrichment of actomyosin and squamous cell morphologies in the OC. Together, our findings demonstrate that intrinsic and extrinsic factors intersect to control actomyosin organization in OC cardiomyocytes, which in turn promotes the cell shape changes that accompany curvature morphogenesis.

Organs are replete with curvature, from the branching distal tips of the lung epithelium to the concave cup of the retina, and to the gyri and sulci of the brain. Essentially every organ exhibits some degree of curvature during its morphogenesis, and the precise curvature geometry required for a particular organ's function is likely sculpted from inputs both extrinsic and intrinsic to its tissues[1]. Decades of interest in this topic show us that some curvatures rely heavily on external influences. For example, the vertebrate gut acquires its dramatic twists and turns due to its rapid lengthening combined with its restrictive attachment to a relatively static partner, the dorsal mesentery[2]. By contrast, the ingressing *Drosophila* ventral furrow is controlled largely by tissue-intrinsic processes, with apical constriction driven by actomyosin pulses in the epithelial cells resulting in robust invagination[3–5]. Here, we investigate the factors, both tissue-extrinsic and -intrinsic,

that control the regionalized cell shape changes that accompany chamber curvature formation in the developing heart.

The heart begins as a simple tube but then transforms into a looped organ with ventricular and atrial chambers. This looping process involves coordinated bending and ballooning of myocardial tissue to form the chambers, the twisting of the heart into a slightly helical shape (known as torsion), and the asymmetric displacement of the ventricle and atrium to deliver the chambers to their ultimate positions[6–10]. During formation of the chambers, each acquires a bulging outer curvature (OC) and a recessed inner curvature (IC)[11–13]. In addition to distinctive tissue morphology, these curvatures exhibit disparate physiological features; for example, the OC exhibits higher conduction velocity[14–16] and more trabeculation[17–19] than the IC, whereas the IC demonstrates greater tissue stiffness[20]. While the

[1]Department of Cell and Developmental Biology, School of Biological Sciences, University of California, San Diego, La Jolla, CA, USA. [2]Department of Developmental Genetics, Max Planck Institute for Heart and Lung Research, Bad Nauheim, Germany. ✉e-mail: dyelon@ucsd.edu

differences between the OC and IC have been well characterized, the mechanisms that initially identify these regions and control their subsequent morphogenesis and specialization remain enigmatic.

Previous experimental and modeling studies have proposed cell behaviors such as hypertrophic growth, rearrangement within the tissue, and changes to cell shape as potential contributors to chamber curvature morphogenesis. For example, in chick, localized increases to cardiomyocyte volume in the OC appear to play a significant role in the creation of the ventricular bend[21,22]. Additionally, in both chick and zebrafish, differing cell rearrangement patterns between the curvatures[9,23], as well as regionalized cell shape changes[24,25], likely play into the pronounced expansion of the OC tissue. In zebrafish in particular, the strikingly divergent morphologies that ventricular OC and IC cardiomyocytes attain during curvature formation have suggested that active cell shape change is a critical driver of curvature formation. Specifically, examination of the myocardial surface of the zebrafish heart tube has highlighted that its cardiomyocytes are fairly uniform in both apical surface area and circularity[25]. However, after curvature formation, OC cardiomyocytes have a significantly larger, more elongated apical surface area than do IC cardiomyocytes[25]. Consistent with the idea that regionalized cell shape changes create chamber curvatures, many mutations that cause chamber dysmorphia also disrupt the acquisition of OC- or IC-linked cell shapes[14,26–30]. However, the subcellular mechanisms that drive these changes to cardiomyocyte morphology have not been well defined.

The actomyosin cytoskeleton is an attractive candidate for mediating cell shape changes. Although capable of inducing shape change in several ways, the network is perhaps best known for its ability to contract and thus create localized tension within a cell, as in the apical constriction observed during *Drosophila* ventral furrow formation[5,31]. This type of actomyosin-derived tension is indeed employed at later stages of zebrafish cardiac morphogenesis, when differential tensions between cardiomyocytes are important for the decision to either trabeculate or remain in the compact layer[32]. If actomyosin also plays an earlier role in curvature cell shape acquisition, we would expect to see different presentations of the network between curvatures. Indeed, in zebrafish, where there is a single layer of cardiomyocytes at this stage, visualization of the outer surface of the myocardium has highlighted what is interpreted as "cortical" F-actin in both curvatures (i.e., intense F-actin signal outlining each cardiomyocyte) and an additional pool of "cytoplasmic" F-actin running throughout OC cardiomyocytes[33]. Although visualizations of the F-actin cytoskeleton in chick are somewhat more challenging due to the bilayered myocardium, distinctions between the OC and IC have been resolved, with circumferentially aligned F-actin at cell-cell boundaries in the IC and a less organized network of F-actin in the OC[34,35]. Furthermore, actin dynamics appear to be necessary for the curvature formation process: in both chick and zebrafish, explanted hearts exposed to inhibitors of actin polymerization fail to achieve the stereotypic curved chamber contours[36–38]. Altogether, these data position actomyosin as a likely regulator of curvature formation, but how the divergence of the actomyosin landscape is regulated and whether the network mediates its effect on tissue shape via active cell shape changes remain unknown.

Curvature formation takes place in a heart that actively beats and pumps blood, suggesting the possible involvement of biomechanical cues in regulating the cellular and subcellular changes that occur in OC and IC cardiomyocytes. It is well known that biomechanical inputs, such as fluid forces stemming from blood flowing through the endocardial lumen, tissue tension arising from the sarcomeric contraction of myocardial cells, and the anchoring of venous and arterial poles to the vascular system, influence several aspects of cardiac development[12,39–44]. For example, altering the dynamics of blood flow in zebrafish can prevent both trabeculation[19,45,46] and valve formation[47–49]. In addition, zebrafish hearts that lack atrial contractility

and thus have reduced blood flow through the ventricle exhibit OC cardiomyocytes with a reduced apical surface area[25]. This phenotype highlights the importance of extrinsic factors for attaining curvature-linked cell shapes, but it is not yet clear how these forces are translated into the subcellular events that instigate changes to cardiomyocyte morphology.

While extrinsic forces clearly impact curvature formation, factors intrinsic to the myocardium are certainly also involved in regulating the cytoskeleton, changing cell behavior, and ultimately sculpting tissue shape. In considering intrinsic factors that could influence curvature morphogenesis, the TBX family of transcription factors stands out as potentially important contributor. A set of these factors work in concert to pattern the expression of genes associated with the chamber curvatures: TBX5 promotes expression of OC-related genes, whereas TBX2 and TBX3 suppress expression of these genes[9,10,50–56]. For example, the combination of these three factors determines the OC-enriched expression of *Nppa* in mice[51–54]. In zebrafish, expression of *nppa* is already enriched in the future ventricular OC region of the linear heart tube, well before curvatures take shape[25]. Although loss-of-function analyses indicate that *nppa* itself is not necessary for curvature formation[57], its early prepattern signifies that there may be other factors regulated by Tbx factors that could influence the ability of OC or IC cardiomyocytes to change their morphology. This idea is especially intriguing in light of recent work in mouse showing that TBX5 influences epithelial tension and cell morphology in the second heart field[58].

In this study, we delve into new aspects of how OC and IC cardiomyocyte morphologies diverge during curvature formation in zebrafish. Specifically, by studying OC and IC cardiomyocyte shapes in three dimensions, we have found that OC cardiomyocytes expand primarily in the planar axis, becoming more squamous as curvatures form, while IC cardiomyocytes extend primarily in the apicobasal axis, becoming more cuboidal. We show that these cellular changes are preceded by regionalized patterns of subcellular actomyosin organization, and that actomyosin plays a cell-intrinsic role in determining cardiomyocyte cell shape. Intriguingly, this role appears to be in the induction of planar spread, rather than the role of constriction for which the actomyosin cytoskeleton is so well known. Finally, we show that blood flow through the ventricle and *tbx5a* each promote the curvature-associated divergence of actomyosin organization and cardiomyocyte morphologies, with particularly prominent effects in the OC. From these findings, we propose that several extrinsic and intrinsic factors converge to control the cytoskeletal dynamics that govern cell shape changes, and that these changes to cardiomyocyte morphologies are a critical component in sculpting the characteristic contours of the cardiac chambers.

## Results

### Curvature formation coincides with the divergence of ventricular OC and IC cardiomyocyte morphologies

To study the cellular and molecular underpinnings of curvature formation, we first needed to develop a working method to standardize the boundaries of the ventricular curvatures. For this purpose, we used a combination of gene expression patterns and measurements of morphological features to guide boundary placement in the embryonic zebrafish ventricle at 48 hours post-fertilization (hpf) (Supplementary Fig. 1a–g). For gene expression, we relied on *nppa*, which is enriched in the OC of the ventricle[25,59]. Generally, the highest *nppa*-expressing area of the ventricle was deemed to be the OC, and the lowest *nppa*-expressing area was deemed to be the IC (Supplementary Fig. 1a, b). After examining the *nppa* expression pattern in 12 wild-type ventricles at 48 hpf, we used the typical territory of *nppa* expression and its spatial relation to the atrioventricular canal (AVC) and distal end of the outflow tract (OFT) to determine the morphological features of the heart that could serve as boundaries for the curvatures

(Supplementary Fig. 1e; see Methods for details). Once we had drawn these boundaries, we found that the IC region coincided satisfyingly with the expression of *mb* (Supplementary Fig. 1c, d), a gene previously noted to be enriched in the ventricular IC[60]. In a wild-type heart at 48 hpf, the OC region defined by this method typically includes ~50–60 cardiomyocytes, and the IC region typically includes ~25–30 cardiomyocytes. We modified this method slightly to determine the curvature boundaries in wild-type hearts at 36 hpf (Supplementary Fig. 1h; see Methods for details), and we used these rules to bound the curvatures in mutant hearts as well (see subsequent figures, including Figs. 4–7). Although we think that these boundaries encompass the territories that are important for the study of ventricular curvature formation, it is possible that they also include some less relevant areas. Nevertheless, this method provides a useful and reproducible strategy for outlining the ventricular OC and IC in multiple contexts.

Our previous work has shown that certain characteristics of cardiomyocyte shape, such as apical surface area and circularity, are uniform throughout the myocardium at the linear heart tube stage (24 hpf), but eventually diverge in the ventricular curvatures by 48 hpf[25]. Employing our new definitions of the OC and IC, we aimed to understand this divergence with greater temporal detail by looking at a developmental stage midway through curvature formation, in both the planar (X/Y) and apicobasal (Z) axes. At 37 hpf, when curvatures have just started to take shape, we found that OC and IC cardiomyocytes exhibit only a modest difference in morphologies (Fig. 1a–f, m–p). OC cardiomyocytes have a slightly but significantly larger apical surface area than IC cardiomyocytes, whereas the circularity of the apical surfaces are comparable (Fig. 1a–c, m, p). Additionally, cell thickness, as measured by apicobasal length, is similar between OC and IC cardiomyocytes at this stage (Fig. 1d–f, n). Both OC and IC cardiomyocytes increase in volume between 37 and 48 hpf and extend this new volume in both planar and apicobasal axes (Fig. 1g–l, o). However, OC cardiomyocytes expand along their planar axis to a greater extent than do IC cardiomyocytes (Fig. 1a–c, g–i), as observed by a greater increase in apical surface area (Fig. 1m), and they become more elongated, typically along their circumferential axis (Fig. 1g–i, p). By contrast, IC cardiomyocytes become taller along their apicobasal axis and remain more circular than OC cells (Fig. 1d–f, j–l, n, p). Together, these data suggest that cardiomyocytes in the developing curvatures grow to similar degrees during this time, but they choose to allocate their new volume along different axes: along the planar axis for OC cardiomyocytes and along the apicobasal axis for IC cardiomyocytes. As a consequence, OC cardiomyocytes become more squamous, and IC cardiomyocytes become more cuboidal. These data also place the onset of cardiomyocyte shape change at an early stage when the heart is still relatively tubular, highlighting active cell shape change as a potential driver of curvature formation.

## Divergence of cardiomyocyte morphologies is preceded by changes in actomyosin organization

To understand the cellular mechanisms behind the divergence in OC and IC cardiomyocyte shapes, we first wanted to identify when previously reported cytoskeletal characteristics of the curvatures diverged[33–35]. Specifically, do the portions of the ventricle that will become the IC and OC exhibit differential actomyosin landscapes as soon as the linear heart tube is formed, or do they diverge sometime during curvature formation? If the latter is true, does this divergence precede the appearance of the observed differences between OC and IC cell shapes? Throughout the process of curvature formation, we found that F-actin primarily resides along the inner surface of the cardiomyocyte membranes. At 28 hpf, when a small bulge of OC is just becoming evident (Supplementary Fig. 3a), the subcellular distribution of F-actin appears fairly uniform in the presumptive OC and IC

(Supplementary Fig. 3b–d). As observed previously[61], proximal ventricular cardiomyocytes (those closer to the AVC) exhibit F-actin mostly at their basal membrane (Supplementary Fig. 3d), whereas distal ventricular cardiomyocytes (those closer to the OFT) exhibit F-actin distributed around all of the surfaces of their membrane, with particularly strong presence at the apical surface (Supplementary Fig. 3c). At 36 hpf, when OC and IC cardiomyocytes have begun to show small morphological differences (Fig. 1a–f, m–p and Supplementary Fig. 3e), the F-actin in OC cardiomyocytes remains more enriched at the basal surface, particularly in the proximal ventricle (Supplementary Fig. 3f–h). In IC cells, F-actin is distributed more equally around the membrane surfaces, with a substantial amount at the lateral and apical surfaces (Supplementary Fig. 3f–h). At 50 hpf, OC and IC cardiomyocytes look fairly similar to each other, with F-actin distributed around all membrane surfaces and a particularly intense signal at the lateral membranes (Supplementary Fig. 3i–l). Our observations that OC and IC cardiomyocytes begin with similar F-actin organization but diverge at 36 hpf, only to converge again by 50 hpf, drew our attention to 36 hpf as a stage worthy of more intense study. These findings correspond well with published visualizations of the myocardial surface showing that F-actin organization differs between OC and IC cells at 36 hpf (but not at 24 hpf)[33] and that *myosin vb* mutants, which fail to undergo OC cell planar expansion, exhibit aberrant F-actin organization in OC cells at 36 hpf[62].

To better understand these qualitative findings, we adopted a more quantitative approach to the examination of the subcellular localization of F-actin and phospho-Myosin (pMyosin) in individual cardiomyocytes of the OC and IC (Fig. 2a–e). For every cardiomyocyte analyzed, we quantified signal intensities to reveal what proportion of the total F-actin and pMyosin is localized to its basal, lateral, and apical membranes (Fig. 2b; see Methods for details). Using this approach, we found that OC cardiomyocytes have a higher proportion of basal actomyosin than do IC cardiomyocytes, whereas IC cardiomyocytes have a higher proportion of apical and lateral actomyosin than do OC cardiomyocytes (Fig. 2c–e). In addition, we wondered whether we could observe disparities between cardiomyocytes in the proximal OC/IC and cells in the distal OC/IC, as in Supplementary Fig. 3. Comparing these subsets of cells, we found that proximodistal differences are quite clear in the OC (Supplementary Figs. 4 and 5). In proximal OC cardiomyocytes, F-actin and pMyosin are highly enriched at the basal surface, whereas in the distal OC, cardiomyocytes have a significantly lower proportion of actomyosin at the basal membrane and a higher proportion of actomyosin at the lateral and apical membranes, giving an overall organization more similar to cardiomyocytes of the IC, which do not exhibit striking differences between proximal and distal regions. These data from individual cells (Fig. 2) support what we observed at the tissue level (Supplementary Fig. 3) and highlight a potentially important aspect of cytoskeletal dynamics during a critical period of curvature formation.

Altogether, our observations indicate that, in the linear heart tube, cardiomyocytes in the future OC and IC have similar actomyosin organization, and that the greater disparity lies between the proximal ventricle and the distal ventricle. As chambers begin to form, cardiomyocytes in the OC (particularly those in the proximal region) retain basal enrichment of actomyosin, while cardiomyocytes in the IC accumulate actomyosin at their apical and lateral surfaces. Notably, the process of acquiring these differential cytoskeletal landscapes closely precedes the initiation of the cell shape changes observed in Fig. 1. These data suggest that the actomyosin network in cardiomyocytes is differentially regulated depending upon their location within the ventricle. In addition, these data imply potentially distinct roles for actomyosin in the curvatures, given that the precise location of actomyosin within a cell determines its ability to engage drivers of specific cell behaviors.

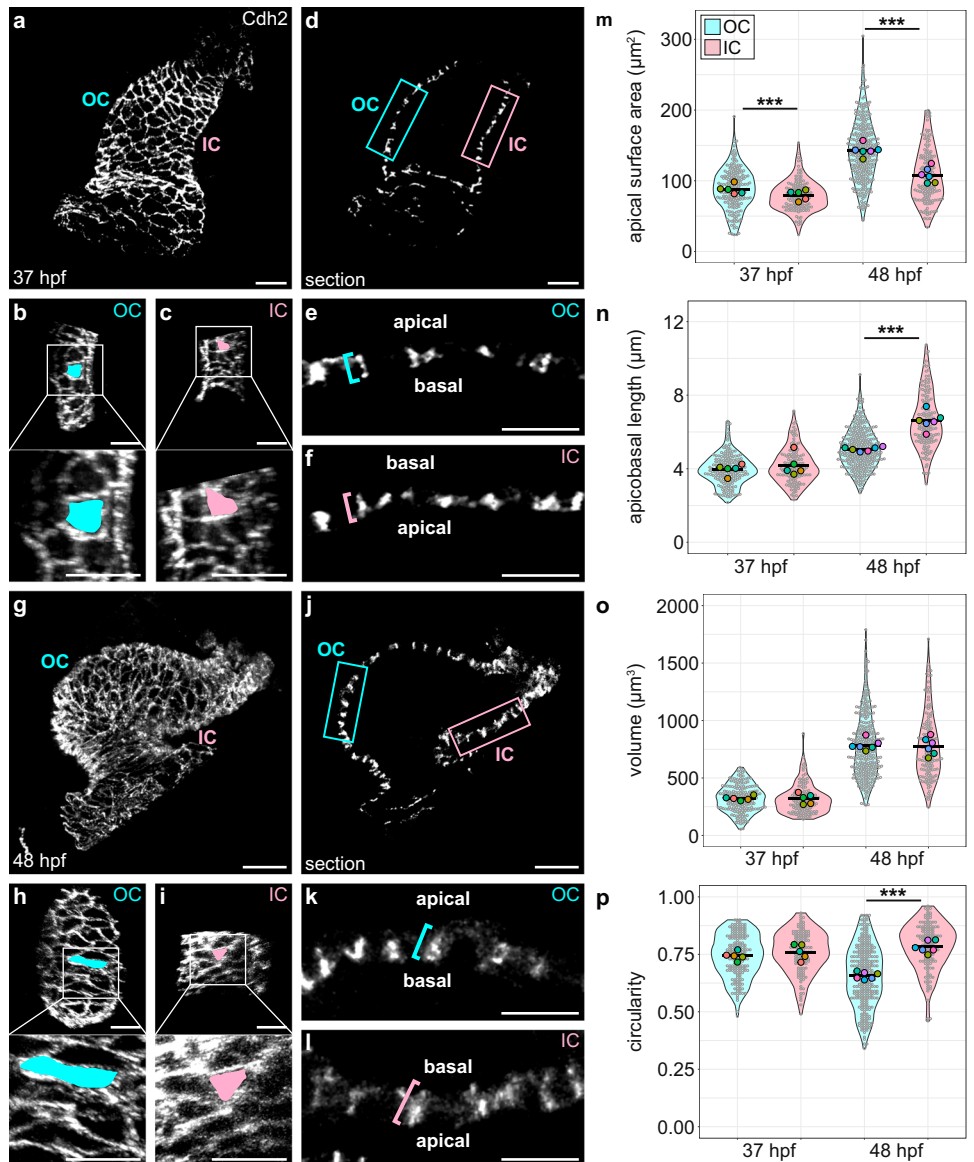

**Fig. 1 | OC and IC cardiomyocyte morphologies diverge during curvature formation.** (**a**, **g**) 3D reconstructions of 37 hpf (**a**) and 48 hpf (**g**) wild-type hearts, dissected from embryos prior to imaging. Immunostaining for Cdh2 labels lateral membranes of cardiomyocytes (see Supplementary Fig. 2 and Methods for more details regarding the use of Cdh2). OC (**b**, **h**) and IC (**c**, **i**) are shown for hearts in (**a**) and (**g**), respectively. Insets show higher magnification. The apical surface area of an individual cardiomyocyte is illustrated by blue or pink fill. **d**, **j** Sections through hearts in (**a**, **g**). (**e**, **f**, **k**, **l**) Magnified views of blue (OC) and pink (IC) boxed regions in (**d**) and (**j**); blue and pink brackets highlight apicobasal length of individual cardiomyocytes. (**m**–**p**) Violin plots compare apical surface area, apicobasal length, volume, or circularity of OC and IC cardiomyocytes at 37 and 48 hpf. Each small gray dot represents an individual cell, each black bar represents the mean of values from individual cells, and each large colored dot represents the mean of all values from an individual embryo. Volume is calculated as LxWxH; circularity is calculated as $4\pi(A/P^2)$. Two-sided Wilcoxon test. **m** 37 hpf OC vs 37 hpf IC: $p = 0.000681$; 48 hpf OC vs 48 hpf IC: $p = 6.26 \times 10^{-15}$. **n** $p = 1.23 \times 10^{-24}$. **p** $p = 1.87 \times 10^{-20}$. Significance only shown for OC/IC comparisons; all metrics are significantly different between developmental stages of the same region (i.e., 37 hpf OC vs 48 hpf OC, and 37 hpf IC vs 48 hpf IC). The data represent two experimental replicates. 37 hpf OC ($N = 5$ embryos, $n = 183$ cells); 37 hpf IC ($N = 5$ embryos, $n = 127$ cells); 48 hpf OC ($N = 6$ embryos, $n = 281$ cells); 48 hpf IC ($N = 6$ embryos, $n = 143$ cells). Scale bars = 30 μm (**a**, **d**, **g**, **j**); 20 μm (**b**, **c**, **h**, **i**); 15 μm (**e**, **f**, **k**, **l**).

## Modulation of actomyosin dynamics dampens the divergence of OC and IC cardiomyocyte morphologies

Given the intriguing regionalized patterns of subcellular actomyosin localization that arise during chamber curvature formation, we wondered whether the actomyosin cytoskeleton plays a role in the divergence of curvature cell shapes. Previous studies in chick and zebrafish have shown that actin polymerization is crucial for chamber curvature formation[36–38], but these studies did not delve into the impact of actin polymerization on cardiomyocyte morphology. We treated wild-type embryos from 24 to 36 hpf or from 36 to 48 hpf with either Latrunculin B (LatB) or Blebbistatin (Bleb) to block actin polymerization or non-

muscle myosin II (NMII) activity, respectively (Supplementary Fig. 6). In embryos treated with either LatB or Bleb from 24 to 36 hpf, OC cardiomyocytes exhibited significantly smaller apical surface areas when compared with DMSO-treated controls (Supplementary Fig. 6a, b, d–f, h). In contrast to the effects of treatment from 24 to 36 hpf, LatB treatment from 36 to 48 hpf had no effect on the planar expansion of OC cardiomyocytes (Supplementary Fig. 6c, d), and, curiously, Bleb treatment from 36 to 48 hpf resulted in excessive planar expansion (Supplementary Fig. 6g, h). These data highlight 24–36 hpf as a particularly important window for actomyosin activity in promoting subsequent OC cardiomyocyte planar expansion. They also hint at a

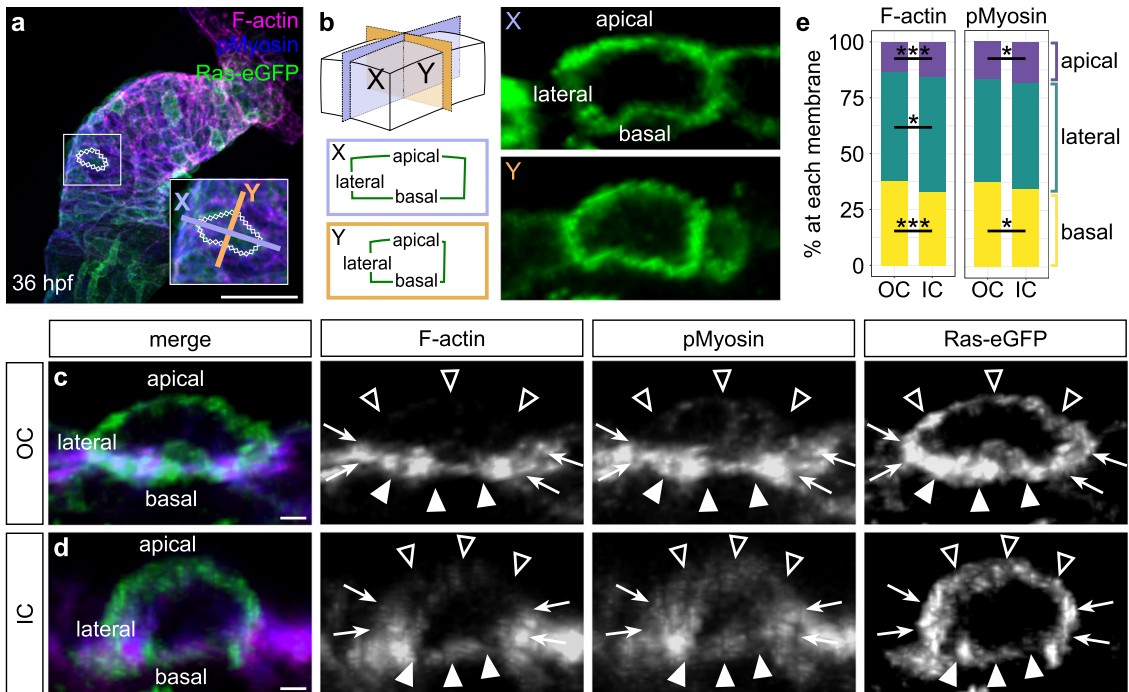

**Fig. 2 | OC and IC cardiomyocytes exhibit differential localization of subcellular actomyosin during early curvature formation. a** Whole heart from a 36 hpf embryo carrying *Tg(myl7:eGFP-Hsa.HRAS)*[97], immunostained for membrane-bound GFP and phospho-Myosin (pMyosin) and stained with Phalloidin to label F-actin. The inset magnifies a single cardiomyocyte. **b** Visual representation of the approach for measuring the percentages of F-actin or pMyosin at each membrane in individual cardiomyocytes. Briefly, each OC or IC cardiomyocyte was bisected in two axes, resulting in two cross-sections for each cell (X, purple; Y, orange). The cell boundaries in X and Y are visible due to eGFP localization to the membranes. In these images, the basal membrane is always at the bottom. **c**, **d** Cross-sections through representative individual cardiomyocytes from the OC (**c**) or IC (**d**). Empty arrowheads: apical membranes. Filled arrowheads: basal membranes. Arrows: lateral membranes. **e** Stacked bar charts showing the mean percentage of F-actin or pMyosin at each membrane. Refer to Supplementary Tables 1 and 2 for summary statistics. Two-sided Wilcoxon test. For F-actin in (**e**), OC vs IC (basal): $p = 9.45 \times 10^{-5}$; OC vs IC (lateral): $p = 0.0148$; OC vs IC (apical): $p = 0.000208$. For pMyosin in (**e**), OC vs IC (basal): $p = 0.0445$; OC vs IC (apical): $p = 0.0212$. The data represent two experimental replicates. OC ($N = 10$ embryos, $n = 199$ cells); IC ($N = 10$ embryos, $n = 123$ cells). Scale bars = 50 μm (**a**); 2 μm (**c**, **d**).

second phase of NMII activity between 36 and 48 hpf, during which Myosin-based contractility might be important for maintaining cardiomyocyte cell shape.

We next wanted to evaluate whether actomyosin activity is required in cardiomyocytes themselves, or if the effect we observed on cardiomyocyte shape upon LatB or Bleb treatment is due to a required function in another tissue, such as the endocardium. To assess these questions, we used established transgenic lines to express two versions of the non-muscle myosin light chain 9 gene (*myl9*) throughout the myocardium, driven by the cardiomyocyte-specific *myl7* promoter, namely: wild-type *myl9* (*Tg(myl7:WT-myl9-mScarlet)*) or a dominant negative version of *myl9* (*Tg(myl7:DN-myl9-eGFP)*)[32]. When assessing cardiomyocyte morphologies at 48 hpf, we found that expression of *WT-myl9* resulted in cardiomyocyte morphologies reminiscent of those in wild-type embryos (Fig. 1g–l, m, n, p and Supplementary Fig. 7a–f, m–o). By contrast, expression of *DN-myl9* resulted in OC cardiomyocytes with decreased apical surface area and increased apicobasal length when compared with OC cardiomyocytes expressing *WT-myl9* (Supplementary Fig. 7g, h, j, k, m, n). Together, these changes resulted in a more cuboidal phenotype. Additionally, IC cardiomyocytes expressing *DN-myl9* exhibited a slightly (but significantly) decreased apical surface area and equivalent apicobasal length when compared with those expressing *WT-myl9* (Supplementary Fig. 7g, i, j, l–n). Thus, since myocardial expression of *DN-myl9* results in OC and IC cardiomyocytes with relatively similar shapes, our data suggest that actomyosin dynamics are indeed important within the myocardium to support the divergence of OC and IC cell morphologies.

To better understand the effect of *DN-myl9* expression on cardiomyocyte shape, we examined whether the changes in NMII activity that led to cell shape alterations at 48 hpf also had an effect on the F-actin network at 36 hpf. Immunostaining for pMyosin verified that the expression of *DN-myl9* reduces NMII activity in the ventricular myocardium (Supplementary Fig. 8a–c), and we found a coincident reduction in myocardial F-actin in *DN-myl9* ventricles (Supplementary Fig. 8a, b, d). By staining with DNase I[63–65], we found that this reduction in F-actin levels was not due to a reduction in available G-actin (Supplementary Fig. 8e–i). Together, these data confirm the efficacy of the *Tg(myl7:DN-myl9-eGFP)* transgene and suggest that NMII activity may be involved in polymerizing actin monomers into F-actin. We further found that subcellular localization of F-actin was disrupted in OC cardiomyocytes expressing *DN-myl9* (Supplementary Fig. 9). Of particular interest, we found a shift in F-actin enrichment from the basal membrane to the apical and lateral membranes in *DN-myl9*-expressing OC cardiomyocytes (Supplementary Fig. 9a, b, e). This shift prompted us to quantify the ratio of the amount of F-actin at the basal membrane to the amount of F-actin at the apical and lateral membranes (basal/(apical + lateral)). This ratio revealed a striking shift away from the basal enrichment normally seen in *WT-myl9*-expressing OC cardiomyocytes (Supplementary Fig. 9f). From other contexts, we know that Myosin itself can alter F-actin characteristics. For example, it can support F-actin bundles by crosslinking[66–68], can induce disassembly of stress fibers[69–71], and can organize F-actin at specific locations within the cell[72–74]. We posit that altering the function of the Myosin network by expression of *DN-myl9* impacts both the abundance and organization of F-actin.

## NMII function plays a cell-intrinsic role in promoting divergence of OC and IC cardiomyocyte morphologies

Having seen that cardiomyocytes fail to attain expected shapes when a dominant negative form of NMII is expressed throughout the myocardium (Supplementary Fig. 7), we were curious about whether actomyosin plays a cell-intrinsic role in promoting OC and IC cell shape divergence, or, rather, if NMII-driven tissue-level tensions are responsible for the acquisition of cell morphologies. We therefore performed blastomere transplantation to create genetically mosaic embryos (Fig. 3a). In these experiments, we analyzed hearts that contained donor-derived cardiomyocytes expressing *DN-myl9* surrounded by host-derived wild-type cardiomyocytes (Fig. 3b–l; see Methods regarding how specific cardiomyocytes were chosen for analysis). In comparison with their wild-type host-derived neighbors, *DN-myl9*-expressing OC cardiomyocytes had a smaller apical surface area when compared with wild-type cardiomyocytes (Fig. 3b–e, j), but apicobasal length was unchanged (Fig. 3c, f, k). This OC cell phenotype is fairly similar to the effect of uniform expression of *DN-myl9* throughout the myocardium (Supplementary Fig. 7g, h, k, m–o). To complement our blastomere transplantation approach, we also injected the *Tg(myl7:DN-myl9-mScarlet)* transgene plasmid into wild-type embryos at the one-cell stage and found reduced apical surface area in OC cardiomyocytes mosaically expressing the transgene when compared with neighboring cardiomyocytes with no detectable *DN-myl9* expression (Supplementary Fig. 10).

In contrast to the effect of modulating NMII activity in OC cardiomyocytes, we found that *DN-myl9*-expressing donor-derived cardiomyocytes in the IC were very similar to their host-derived neighbors (Fig. 3g–l). As in the OC, this finding mirrors the effect of uniform expression of *DN-myl9* throughout the myocardium (Supplementary Fig. 7g, i, l, m–o). Taken together, our analysis of cardiomyocytes expressing *DN-myl9* emphasizes that actomyosin activity supports the divergence of OC and IC cell shapes by promoting the squamous identity of OC cardiomyocytes in a cell-intrinsic manner, perhaps by facilitating active expansion in the planar axis, as has been hypothesized[37]. In contrast, IC cardiomyocytes appear to be comparatively unaffected by reduced NMII activity.

## Divergence of OC and IC actomyosin organization and cardiomyocyte morphology depends upon biomechanical cues

Our results thus far had highlighted actomyosin organization and function as drivers of cell shape change during curvature formation, and we next wanted to know what types of upstream regulators might control these subcellular features. We were particularly curious to evaluate how blood flow might affect actomyosin localization and cardiomyocyte morphology, as our previous work has shown that myosin heavy chain 6 (*myh6*) mutants[75], which lack an atrium-specific myosin heavy chain and therefore lack atrial contractility, have reduced blood flow through the ventricle as well as reduced apical surface area of OC cardiomyocytes[25]. Consistent with previous findings, we found that OC cardiomyocytes of *myh6* mutants fail to extend along the planar axis (Fig. 4a, b, g, h, m and Supplementary Fig. 11). In addition, we observed that *myh6* OC cardiomyocytes also extend inappropriately along the apicobasal axis (Fig. 4d, e, j, k, n and Supplementary Fig. 11). In contrast to the OC results, we found that the apical surface area and apicobasal length of *myh6* IC cardiomyocytes are indistinguishable from those of wild-type IC cells (Fig. 4a, c, d, f, g, i, j, l–n and Supplementary Fig. 11); however, both OC and IC cells exhibit a modest but significant decrease in volume in *myh6* mutants (Fig. 4o).

Given the effect of reduced blood flow on OC cell morphology, we next asked if this phenotype is coupled with a dysregulated actomyosin cytoskeleton. We found that the overall levels of pMyosin and F-actin in the ventricular myocardium in *myh6* mutants at 36 hpf are reduced relative to wild-type (Supplementary Fig. 12a–d), similar to our observations in *DN-myl9* transgenic hearts (Supplementary Fig.

8a–d). Also resembling the *DN-myl9* scenario (Supplementary Fig. 8e–i), *myh6* mutant hearts exhibit levels of G-actin similar to wild-type (Supplementary Fig. 12e–i), suggesting that the F-actin deficiency is due to a polymerization defect rather than limited availability of actin monomers. In addition to an overall deficit in F-actin, we found mislocalization of subcellular actomyosin in *myh6* mutant OC cardiomyocytes (Fig. 5a–d, g, h): both F-actin and pMyosin shift away from the basal surface and to the lateral and apical surfaces in the proximal OC and distal OC, with a particularly striking shift in the proximal OC (Fig. 5c, d, g, h and Supplementary Fig. 12j). In contrast to this OC phenotype, both F-actin and pMyosin organization in *myh6* mutant IC cardiomyocytes are comparable to wild-type (Fig. 5e–h and Supplementary Fig. 12j). Taken together, these data highlight a previously unappreciated role for extrinsic factors such as the biomechanical cues produced by blood flow in ensuring the regionalization of cytoskeletal landscapes and, subsequently, the planar expansion of OC cardiomyocytes.

## Cell-autonomous function of *tbx5a* contributes to the divergence of OC and IC actomyosin organization and cardiomyocyte morphology

Alongside considering the extrinsic factors that influence curvature formation, we also wondered what types of intrinsic factors, operating within individual cells, could impact the divergence of cardiomyocyte cell shapes and actomyosin organization. We were particularly curious about the influence of the T-box transcription factor gene *tbx5a*. In addition to the known role for Tbx5 in promoting OC transcriptional programs[50,51,55], zebrafish homozygous for mutant alleles of *tbx5a* exhibit a highly dysmorphic heart by 48 hpf[9,76]. As reported previously[76], the outward bulge of the *tbx5a* mutant OC is substantially reduced, and the *tbx5a* mutant IC exhibits less of a kink when compared with a wild-type heart (Fig. 6a, g). We therefore investigated whether *tbx5a* mutant cardiomyocytes underwent the stereotypical divergence in shape associated with the ventricular curvatures. At the cellular scale, and similar to the phenotypes seen in *DN-myl9*-expressing hearts (Supplementary Fig. 7) and in *myh6* mutants (Fig. 4), OC cardiomyocytes in *tbx5a* mutants fail to expand in the planar axis and undergo aberrant apicobasal expansion (Fig. 6a, b, d, e, g, h, j, k, m, n). This combination results in *tbx5a* mutant OC cardiomyocytes with a more cuboidal phenotype than wild-type OC cardiomyocytes (Supplementary Fig. 11). Interestingly, even though at the tissue level *tbx5a* mutant ICs fail to form the stereotypic kink (Fig. 6g), IC cell shapes in *tbx5a* mutants appear comparable to wild-type with just a slight trend toward reduction in apicobasal length (Fig. 6a, c, d, f, g, i, j, l–n and Supplementary Fig. 11). In contrast to *myh6* mutants (Fig. 4o, p), *tbx5a* OC and IC cardiomyocytes exhibit volumes and circularity similar to that of wild-type cardiomyocytes (Fig. 6o, p).

We next wondered whether the effect of *tbx5a* loss-of-function on OC cardiomyocyte morphology was preceded by a dysregulation of the actomyosin cytoskeleton. Similar to the *DN-myl9* (Supplementary Fig. 8a, b, d–i) and *myh6* (Supplementary Fig. 12a, b, d–i) phenotypes, the overall levels of F-actin in the *tbx5a* mutant ventricular myocardium are reduced at 36 hpf, even though G-actin levels appear similar to those of wild-type (Supplementary Fig. 13a, b, d–i). In contrast to the *DN-myl9* and *myh6* phenotypes (Supplementary Figs. 8a–c and 12a–c), however, levels of pMyosin in *tbx5a* mutants are not significantly different from wild-type (Supplementary Fig. 13a–c), suggesting that Tbx5a-regulated pathways affect actin polymerization independently of NMII activity. In addition to an overall deficit in F-actin, we found mislocalization of subcellular F-actin and pMyosin in *tbx5a* mutants, and this was particularly prominent in the proximal OC (Fig. 7a–h). In the *tbx5a* mutant proximal OC, F-actin and pMyosin enrichment shift away from the basal surface and accumulate more at the lateral and apical surfaces (Fig. 7g, h and Supplementary Fig. 13j). In distal OC and IC cardiomyocytes, F-actin is also shifted laterally and

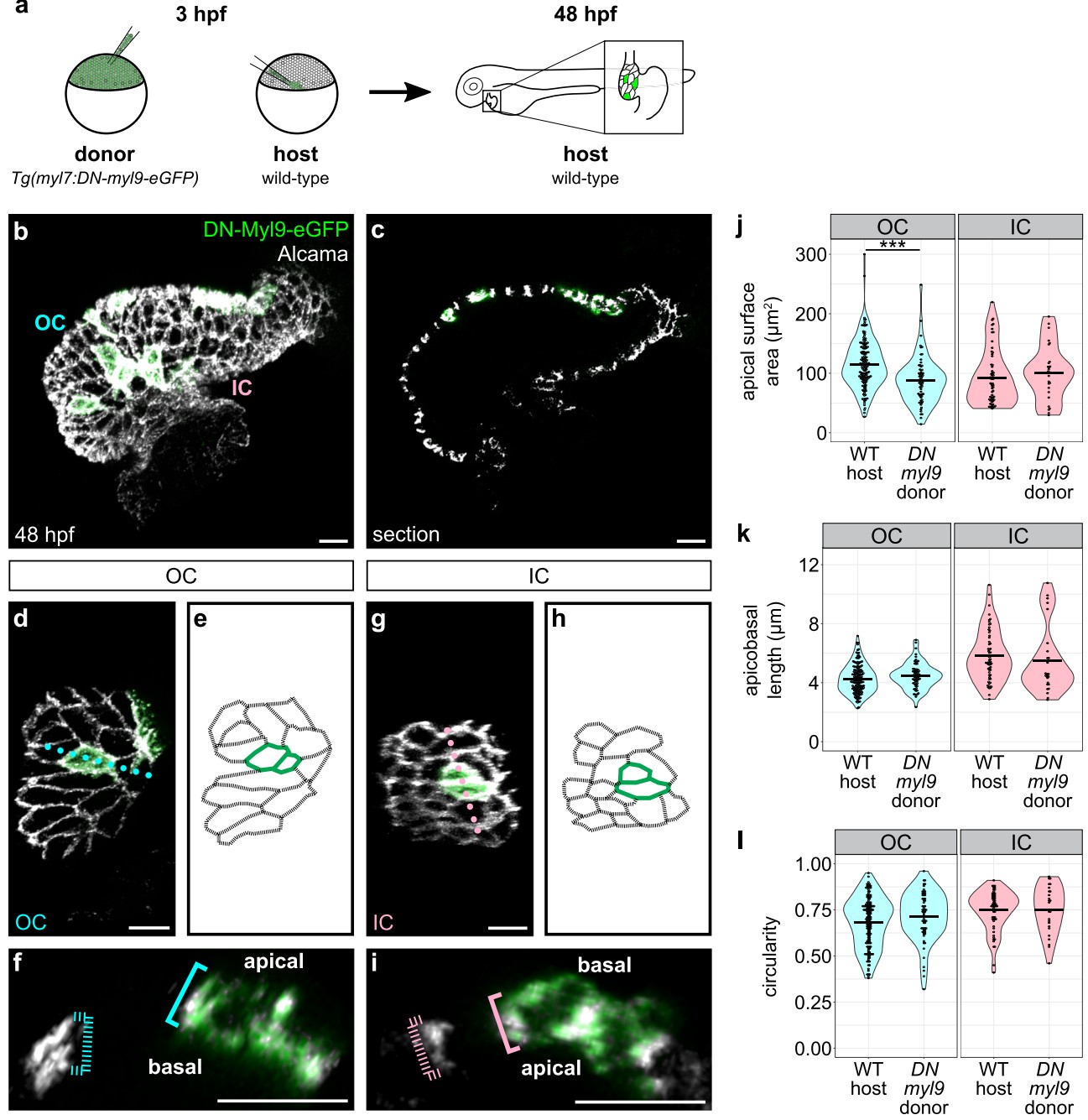

**Fig. 3 | Cell-intrinsic inhibition of NMII activity restricts planar expansion in OC cardiomyocytes. a** Schematic of blastomere transplantation experiment. **b** 3D reconstruction of a 48 hpf wild-type (WT) host heart containing donor-derived cardiomyocytes expressing *Tg(myl7:DN-myl9-eGFP)*; immunostaining for Alcama labels lateral membranes of cardiomyocytes. **c** Section through the heart in (**b**). **d, g** OC (**d**) and IC (**g**) are shown from additional mosaic hearts other than that in (**b**). **e, h** Tracings of the cardiomyocytes in (**d**) and (**g**). Green outlines indicate donor-derived cardiomyocytes; black outlines indicate host-derived cardiomyocytes. **f, i** Cross-sections through positions indicated by dotted lines in (**d**) and (**g**). Blue and pink brackets highlight apicobasal length of individual cardiomyocytes, with dashed brackets for host-derived cardiomyocytes and solid brackets for donor-derived cardiomyocytes. **j–l** Violin plots compare apical surface area, apicobasal length, and circularity of host-derived cardiomyocytes to those of donor-derived cardiomyocytes. Each dot represents an individual cell. Two-sided Wilcoxon test. **j** $p = 1.04 \times 10^{-6}$. Data gathered from 4 days of transplantation. Host OC ($N = 10$ embryos, $n = 156$ cells); donor OC ($N = 10$ embryos, $n = 65$ cells); host IC ($N = 8$ embryos, $n = 60$ cells); donor IC ($N = 8$ embryos, $n = 27$ cells). Scale bars = 15 μm.

apically (albeit to a lesser extent than in the proximal OC), but the ratio of basal to apical/lateral pMyosin is comparable to wild-type (Fig. 7g, h and Supplementary Fig. 13j). Intriguingly, in contrast to the *myh6* mutant scenario where the effect on pMyosin localization is very similar to the effect on F-actin localization (Fig. 5g, h and Supplementary Fig. 12j), the F-actin phenotype observed in *tbx5a* mutants is more severe than the pMyosin phenotype (Fig. 7g, h and

Supplementary Fig. 13j). Overall, these data suggest that Tbx5a-regulated pathways are particularly important for the enrichment of F-actin at the basal surface of OC cardiomyocytes.

Given the actomyosin and cell morphology phenotypes in flow-deficient *myh6* mutants (Figs. 4 and 5) and the reduced heart rate in *tbx5a* mutants[76], we wondered whether the aberrant effects on F-actin localization in *tbx5a* mutants reflect indirect consequences of altered

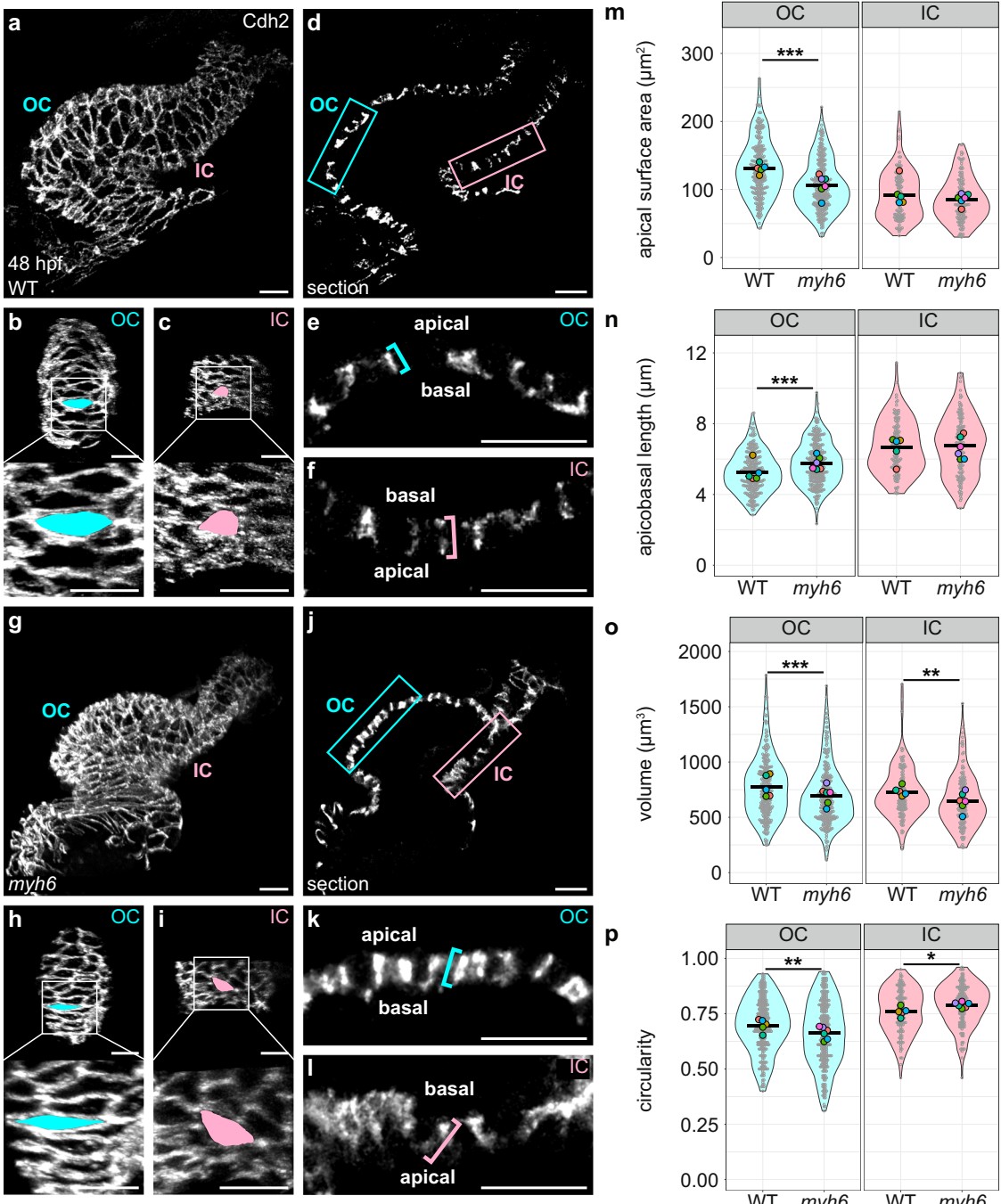

**Fig. 4 | Reduced blood flow inhibits the divergence of OC and IC cardiomyocyte shapes. a, g** 3D reconstructions of wild-type (**a**) and *myh6* mutant (**g**) hearts at 48 hpf; immunostaining for Cdh2 labels lateral membranes of cardiomyocytes. OCs (**b**, **h**) and ICs (**c**, **i**) are shown for hearts in (**a**, **g**). Insets show higher magnification. The apical surface area of an individual cardiomyocyte is illustrated by blue or pink fill. **d, j** Sections through hearts in (**a**, **g**). (**e, f, k, l**) Magnified views of blue (OC) and pink (IC) boxed regions in (**d**, **j**); blue and pink brackets highlight apicobasal length of individual cells. **m–p** Violin plots compare apical surface area, apicobasal length, volume, and circularity for cells from wild-type and *myh6* mutant hearts. **p** Note that *myh6* OC cardiomyocytes are more elongated than wild-type OC cardiomyocytes, a finding that contrasts with our previous work[25]. We posit that this

discrepancy could arise from changes in the genetic background over time or from differences in how data were collected (e.g., where the OC boundary was drawn or how individual cardiomyocytes were measured). For all violin plots, each small gray dot represents an individual cell, each black bar represents the mean of values from individual cells, and each large colored dot represents the mean of all values from an individual embryo. Two-sided Wilcoxon test. **m** $p = 8.97 \times 10^{-13}$. **n** $p = 2.75 \times 10^{-6}$. **o** WT OC vs *myh6* OC: $p = 0.000129$; WT IC vs *myh6* IC: $p = 0.00244$. **p** WT OC vs *myh6* OC: $p = 0.00391$; WT IC vs *myh6* IC: $p = 0.0244$. The data represent one experimental replicate. Wild-type OC ($N = 5$ embryos, 267 cells); *myh6* OC ($N = 6$ embryos, $n = 275$ cells); wild-type IC ($N = 5$ embryos, $n = 120$ cells); *myh6* IC ($N = 6$ embryos, $n = 135$ cells). Scale bars = 20 μm.

blood flow or a cell-intrinsic loss of *tbx5a* function. To address this, we took a mosaic approach in which we transplanted wild-type blastomeres into *tbx5a* mutant host embryos and *tbx5a* mutant blastomeres into wild-type host embryos, as well as performing wild-type

into wild-type and mutant into mutant controls (Fig. 8a–d). We fixed host embryos at 36 hpf and assessed amounts of F-actin at basal, lateral, and apical membranes (Fig. 8e–p and Supplementary Fig. 14). In a *tbx5a* into wild-type scenario, we found that the F-actin organization in

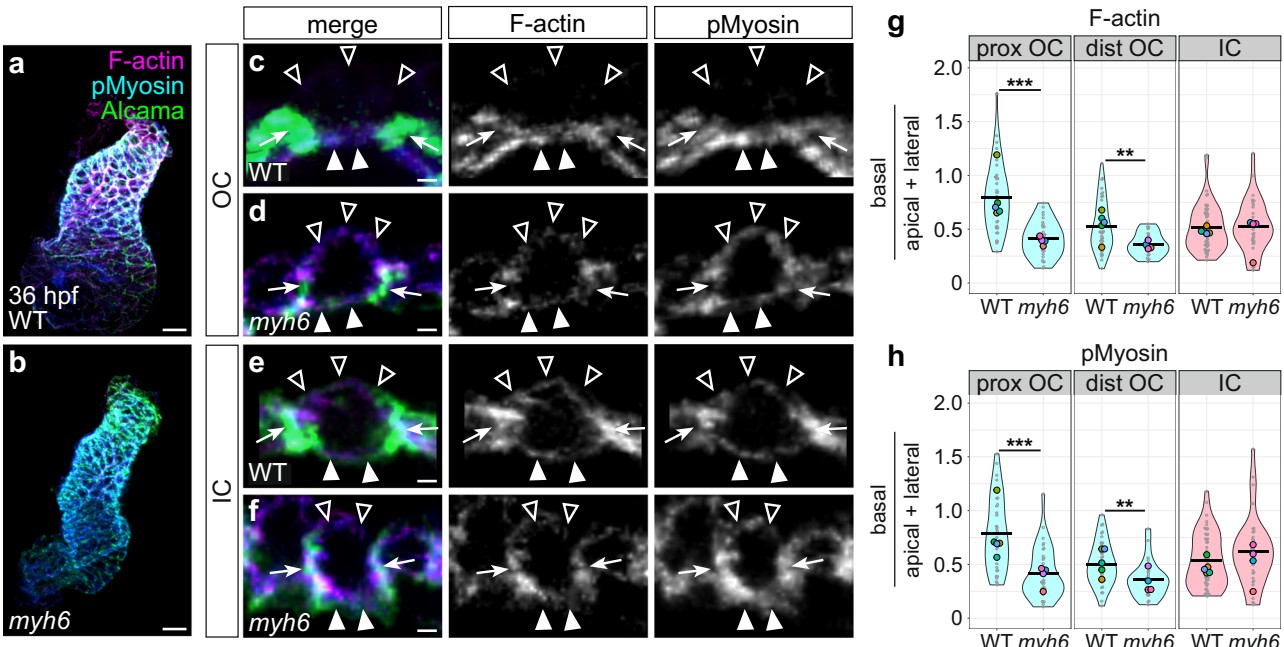

**Fig. 5 | Reduced blood flow inhibits the divergence of OC and IC actomyosin organization. a**, **b** Wild-type (**a**) and *myh6* mutant (**b**) hearts at 36 hpf, immunostained for pMyosin and stained with Phalloidin to label F-actin. Immunostaining for Alcama labels lateral membranes of cardiomyocytes. **c–f** Cross-sections through representative cardiomyocytes from the OC (**c**, **d**) and IC (**e**, **f**) of 36 hpf wild-type (**c**, **e**) and *myh6* mutant (**d**, **f**) hearts. Empty arrowheads: apical membranes. Filled arrowheads: basal membranes. Arrows: lateral membranes. **g** Violin plots show calculated values of (mean basal F-actin/(mean apical F-actin + mean lateral F-actin)) for individual cells. **h** Violin plots of calculated values as in (**g**), but for pMyosin. For all violin plots, each small gray dot represents an individual cell, each

black bar represents the mean of values from individual cells, and each large colored dot represents the mean of all values from an individual embryo. Two-sided Wilcoxon test. **g** WT prox OC vs *myh6* prox OC: $p = 9.48 \times 10^{-9}$; WT dist OC vs *myh6* dist OC: $p = 0.00172$. **h** WT prox OC vs *myh6* prox OC: $p = 5.19 \times 10^{-7}$; WT dist OC vs *myh6* dist OC: $p = 0.00254$. Data represent one experimental replicate. Wild-type proximal OC ($N = 5$ embryos, $n = 39$ cells); *myh6* proximal OC ($N = 4$ embryos, $n = 32$ cells); wild-type distal OC ($N = 5$ embryos, $n = 41$ cells); *myh6* distal OC ($N = 4$ embryos, $n = 24$ cells); wild-type IC ($N = 5$ embryos, $n = 54$ cells); *myh6* IC ($N = 4$ embryos, $n = 35$ cells). Scale bars = 20 µm (**a**, **b**); 2 µm (**c–f**).

*tbx5a* mutant donor-derived proximal OC cardiomyocytes largely resembled that of their wild-type neighbors (Fig. 8k, m and Supplementary Fig. 14c), indicating that something about the wild-type environment can rescue this aspect of the *tbx5a* mutant phenotype. In light of this result, we were particularly intrigued by the wild-type into *tbx5a* scenario. Here, we found that wild-type donor-derived cardiomyocytes in a *tbx5a* mutant proximal OC contained a significantly higher proportion of basal F-actin compared with apical and lateral F-actin, when compared with their *tbx5a* mutant neighbors (Fig. 8h, j and Supplementary Fig. 14b). This suggests an ability of wild-type cardiomyocytes to organize their F-actin network regardless of the *tbx5a* status of surrounding cells. Together, these data indicate that there is at least a partially cell-autonomous role for *tbx5a* in establishing the divergent actomyosin organization observed in the developing curvatures, but that there are also likely to be non-autonomous ways that *tbx5a* influences actomyosin organization.

We next wanted to know whether *tbx5a* also plays a cell-autonomous role in promoting the divergence of OC and IC cardiomyocyte morphologies, or if the cardiomyocyte shape anomalies observed in Fig. 6 are due to cardiac function defects present in *tbx5a* mutants[76]. To distinguish between these possibilities, we used the same experimental scheme as in Fig. 8, but we instead fixed host embryos at 48 hpf and assessed cardiomyocyte shapes (Supplementary Fig. 15). Here, we focused on assessing apical surface area, as our data suggest a primary role for actomyosin in promoting the planar expansion of cardiomyocytes, particularly in the OC. Interestingly, consistent with our analysis of actomyosin localization in wild-type into *tbx5a* mosaic hearts (Fig. 8h, j), wild-type donor-derived OC cardiomyocytes extended in the planar axis significantly more than their *tbx5a* mutant host-derived neighbors, allowing them to assume a more

squamous identity (Supplementary Fig. 15h, j). Also like our mosaic analysis of actomyosin localization (Fig. 8k, m), *tbx5a* mutant donor-derived cardiomyocytes that resided in a wild-type OC largely exhibited morphology similar to their wild-type neighbors (Supplementary Fig. 15k, m). Finally, donor-derived IC cardiomyocytes in both wild-type into mutant and mutant into wild-type scenarios achieved morphologies similar to their host-derived neighbors (Supplementary Fig. 15i, j, l, m). Taking our transplantation experiments together, it appears that the environment of a wild-type heart can largely override the absence of *tbx5a* function in donor-derived mutant cardiomyocytes, but, conversely, wild-type cardiomyocytes can achieve stereotypical OC traits even in a *tbx5a* mutant background. This suggests that the ventricular phenotypes we see in *tbx5a* mutants are not solely due to defects in blood flow and that there is some cell-autonomous aspect to *tbx5a* function during curvature formation.

An unexpected finding from our mosaic studies was that, in some cases, donor-derived cardiomyocytes sent basal projections underneath their host-derived neighbors (Supplementary Fig. 16). We observed such projections in all four of our transplant scenarios; these projections were typically rather short and slim and were usually restricted to the junctions between neighboring cells (Supplementary Fig. 16b, e). However, in the wild-type into *tbx5a* mutant transplant scenario, it was not uncommon for wild-type donor-derived OC cardiomyocytes to send long, broad projections underneath the cell body of a neighboring *tbx5a* mutant host-derived cardiomyocyte (Supplementary Fig. 16c–e). This could be a reflection of the comparatively robust ability of wild-type cardiomyocytes to expand their basal domain, a relatively reduced cell-extracellular matrix (ECM) connection in *tbx5a* mutant tissue, or both. This idea is further supported by the observation that, in the same wild-type into *tbx5a* mutant scenario,

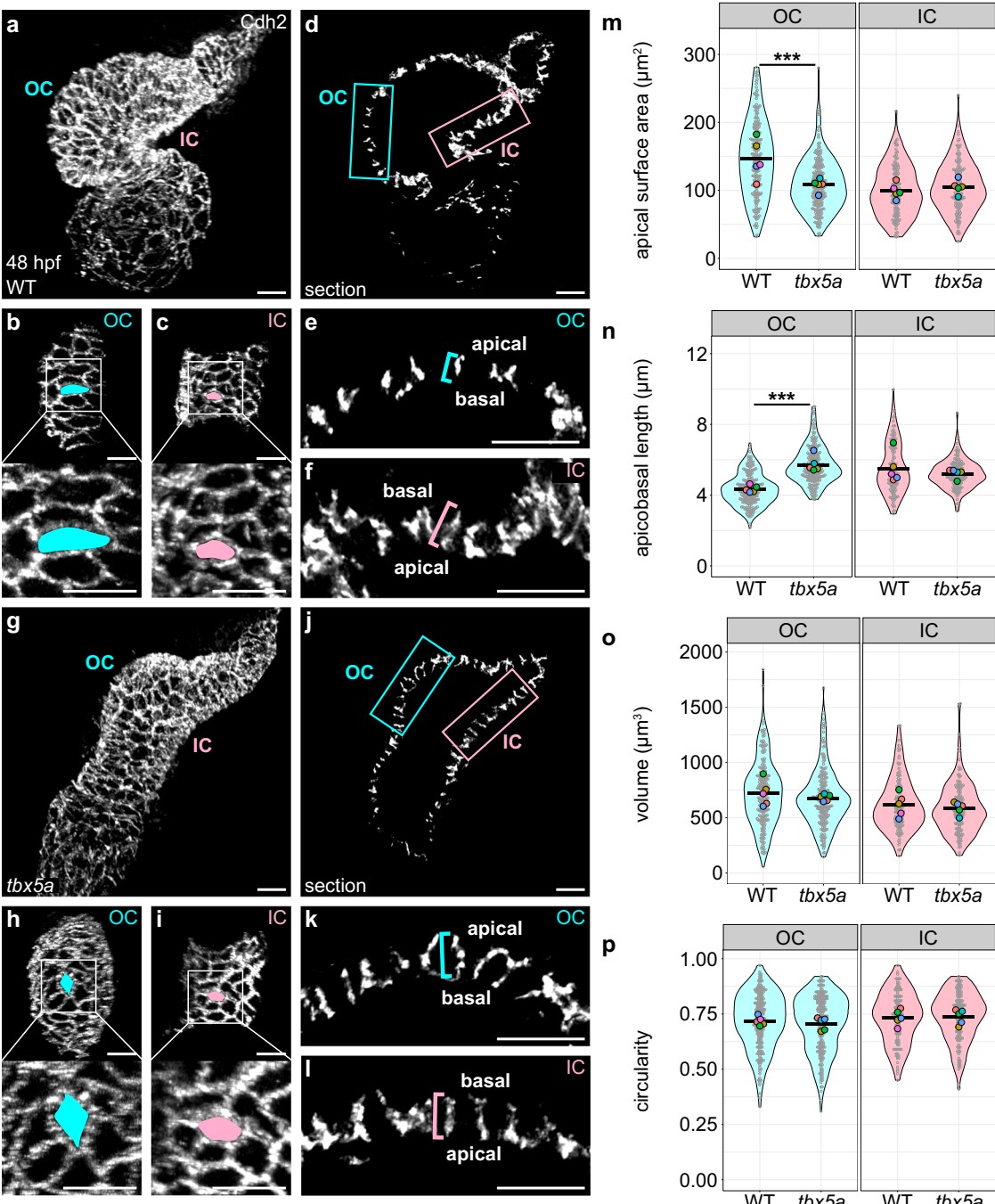

**Fig. 6 | *tbx5a* regulates pathways that support the divergence of OC and IC cardiomyocyte morphologies. a**, **g** 3D reconstructions of wild-type (**a**) and *tbx5a* mutant (**g**) hearts at 48 hpf; immunostaining for Cdh2 labels lateral membranes of cardiomyocytes. OCs (**b**, **h**) and ICs (**c**, **i**) are shown for hearts in (**a**, **g**). Insets show higher magnification. The apical surface area of an individual cardiomyocyte is illustrated by blue or pink fill. **d**, **j** Sections through hearts in (**a**, **g**). **e**, **f**, **k**, **l** Magnified views of blue (OC) and pink (IC) boxed regions in (**d**, **j**); blue and pink brackets highlight apicobasal length of individual cardiomyocytes. **m**–**p** Violin plots compare apical surface area, apicobasal length, volume, and circularity for cardiomyocytes from wild-type and *tbx5a* mutant hearts. For all violin plots, each small gray dot represents an individual cell, each black bar represents the mean of values from individual cells, and each large colored dot represents the mean of all values from an individual embryo. Two-sided Wilcoxon test. **m** $p = 1.03 \times 10^{-13}$. **n** $p = 1.39 \times 10^{-35}$. The data represent one experimental replicate. Wild-type OC ($N = 5$ embryos, $n = 214$ cells); *tbx5a* OC ($N = 5$ embryos, $n = 220$ cells); wild-type IC ($N = 5$ embryos, $n = 126$ cells); *tbx5a* IC ($N = 5$ embryos, $n = 141$ cells). Scale bars = 20 μm.

donor-derived OC cardiomyocytes that did not contact any other donor-derived OC cardiomyocytes (i.e., those in single-cell clones) had, on average, a higher apical surface area than donor-derived OC cardiomyocytes that contacted other donor-derived cardiomyocytes (Supplementary Fig. 17a). Further morphological assessment of these single-cell clones showed that all of them sent out basal projections:

three of the seven clones sent thin basal projections that extended between the junctions of neighboring host cells (Supplementary Fig. 17b), and the remaining four sent larger basal projections underneath neighboring host-derived cells (Supplementary Fig. 17c). We propose that this ability to send basal projections is a mechanism that normally drives planar expansion of OC cardiomyocytes, and that this activity is

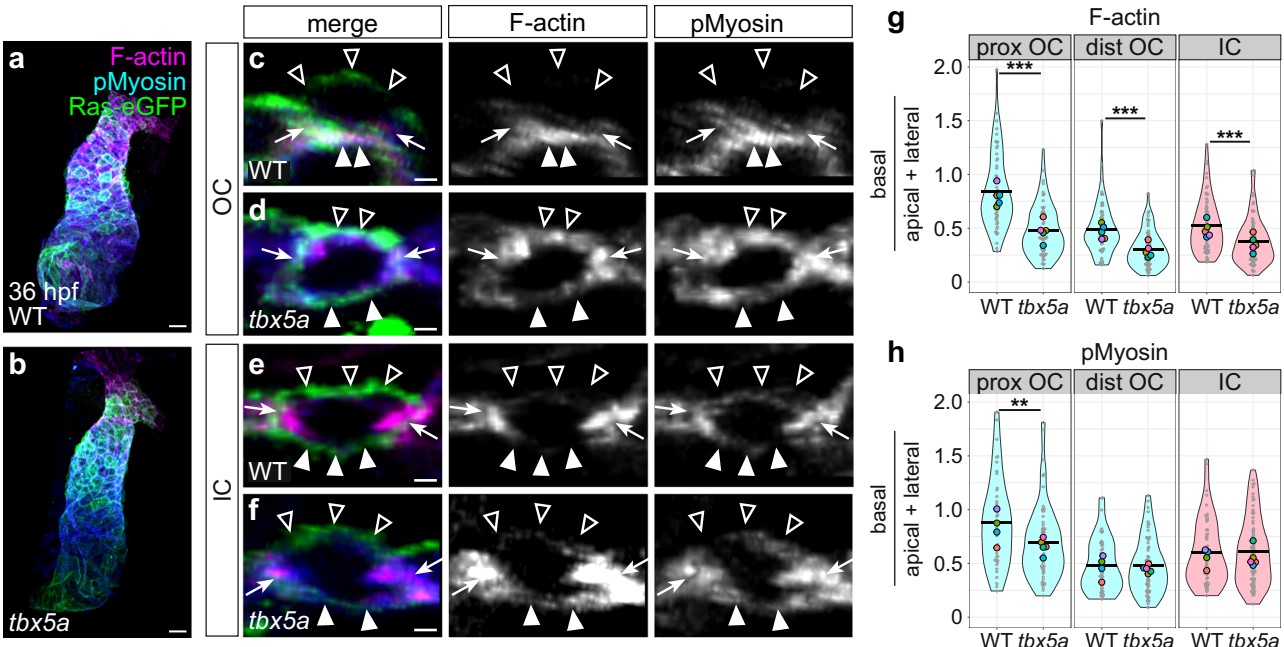

**Fig. 7 | *tbx5a* regulates pathways that support the divergence of OC and IC cardiomyocyte actomyosin organization. a, b** Wild-type (**a**) and *tbx5a* mutant (**b**) hearts at 36 hpf, immunostained for membrane-bound eGFP and pMyosin, and stained with Phalloidin to label F-actin. **c–f** Cross-sections through representative cardiomyocytes from the OC (**c, d**) and IC (**e, f**) of 36 hpf wild-type (**c, e**) and *tbx5a* mutant (**d, f**) hearts. Empty arrowheads: apical membranes. Filled arrowheads: basal membranes. Arrows: lateral membranes. **g** Violin plots show calculated values of (mean basal F-actin/(mean apical F-actin + mean lateral F-actin)) for individual cells. **h** Violin plots of calculated values as in (**g**), but for pMyosin. For all violin plots,

each small gray dot represents an individual cell, each black bar represents the mean of values from individual cells, and each large colored dot represents the mean of all values from an individual embryo. Two-sided Wilcoxon test. **g** WT prox OC vs *tbx5a* prox OC: $p = 5.81 \times 10^{-10}$; WT dist OC vs *tbx5a* dist OC: $p = 9.57 \times 10^{-7}$; WT IC vs *tbx5a* IC: $p = 6.10 \times 10^{-5}$. **h** $p = 0.00958$. The data represent one experimental replicate. Wild-type proximal OC ($N = 5$ embryos, $n = 58$ cells); *tbx5a* proximal OC ($N = 5$ embryos, $n = 64$ cells); wild-type distal OC ($N = 5$ embryos, $n = 61$ cells); *tbx5a* distal OC ($N = 5$ embryos, $n = 67$ cells); wild-type IC ($N = 5$ embryos, $n = 69$ cells); *tbx5a* IC ($N = 5$ embryos, $n = 81$ cells). Scale bars = 20 μm (**a, b**); 2 μm (**c–f**).

increased when neighboring cardiomyocytes maintain a looser attachment to the underlying ECM. Additionally, these data further support a cell-autonomous role for *tbx5a* in directing cardiomyocyte behavior.

## Discussion

Through analysis of the cellular and subcellular dynamics of individual cardiomyocytes within the developing ventricle, we have gained several insights into the mechanisms that drive the morphological divergence of OC and IC cardiomyocytes and the potential contribution of this divergence to chamber curvature formation. Although cardiomyocytes in both curvatures increase in volume during curvature emergence, cardiomyocytes in the OC expand preferentially in the planar axis, whereas cardiomyocytes in the IC expand preferentially in the apicobasal axis. These contrasting behaviors are closely preceded by a divergence in subcellular actomyosin localization between the curvatures. Further, inhibition of actomyosin activity restricts the ability of OC cardiomyocytes to attain the expected squamous shape, indicating that the actomyosin network plays an important role in promoting squamous instead of cuboidal cardiomyocyte morphologies. Finally, the regionalized distinctions between the actomyosin landscapes and, subsequently, the cell morphologies in the curvatures are influenced by both extrinsic factors, such as blood flow through the ventricle, and intrinsic factors, like the transcription factor Tbx5a.

Taken together, our data suggest a novel model in which regional regulation of the actomyosin cytoskeleton mediates the planar expansion of OC cardiomyocytes, ultimately causing a divergence in OC and IC cell morphologies during a critical period of curvature formation (Fig. 9). Prior to the onset of curvature formation, the largest distinctions in actomyosin organization within the ventricle are

primarily proximal versus distal (Supplementary Fig. 3), in agreement with previous studies[61]. As development proceeds, however, differences begin to emerge between the subcellular landscapes of OC and IC cardiomyocytes. Specifically, by 36 hpf, cardiomyocytes throughout the proximodistal length of the IC acquire more apical actomyosin, whereas cardiomyocytes especially in the proximal OC maintain strong basal localization of actomyosin (Fig. 2 and Supplemental Fig. 4). We propose that this basal enrichment of actomyosin promotes subsequent planar expansion of OC cardiomyocytes between 36 and 48 hpf, perhaps by engaging the focal adhesion machinery and components of the underlying cardiac jelly. The coincident lack of actomyosin at the apical surface of proximal OC cardiomyocytes may allow for low localized tension, ensuring that the apical surface can passively expand along with the actively expanding basal surface. In the IC, we speculate that the redistribution of actomyosin away from the basal surface reduces certain associations with the ECM and therefore reduces outward pushing forces; simultaneously, more actomyosin at the apical surface could play a more active and stereotypically contractile role in maintaining a compact apical surface area. Thus, we envision that the actomyosin landscapes in each curvature support OC cardiomyocytes in expanding outward along the planar axis and IC cardiomyocytes in remaining more constricted. Future endeavors to target the cytoskeletal elements in each curvature will help to test specific tenets of this model and to determine whether distinct actomyosin landscapes are indeed causing the changes to cell shape observed here.

This model of contrasting OC and IC cell behaviors is also supported by aspects of our mosaic analyses. For example, we found that wild-type cardiomyocytes in the midst of a *tbx5a* mutant OC often send large, broad projections underneath neighboring mutant

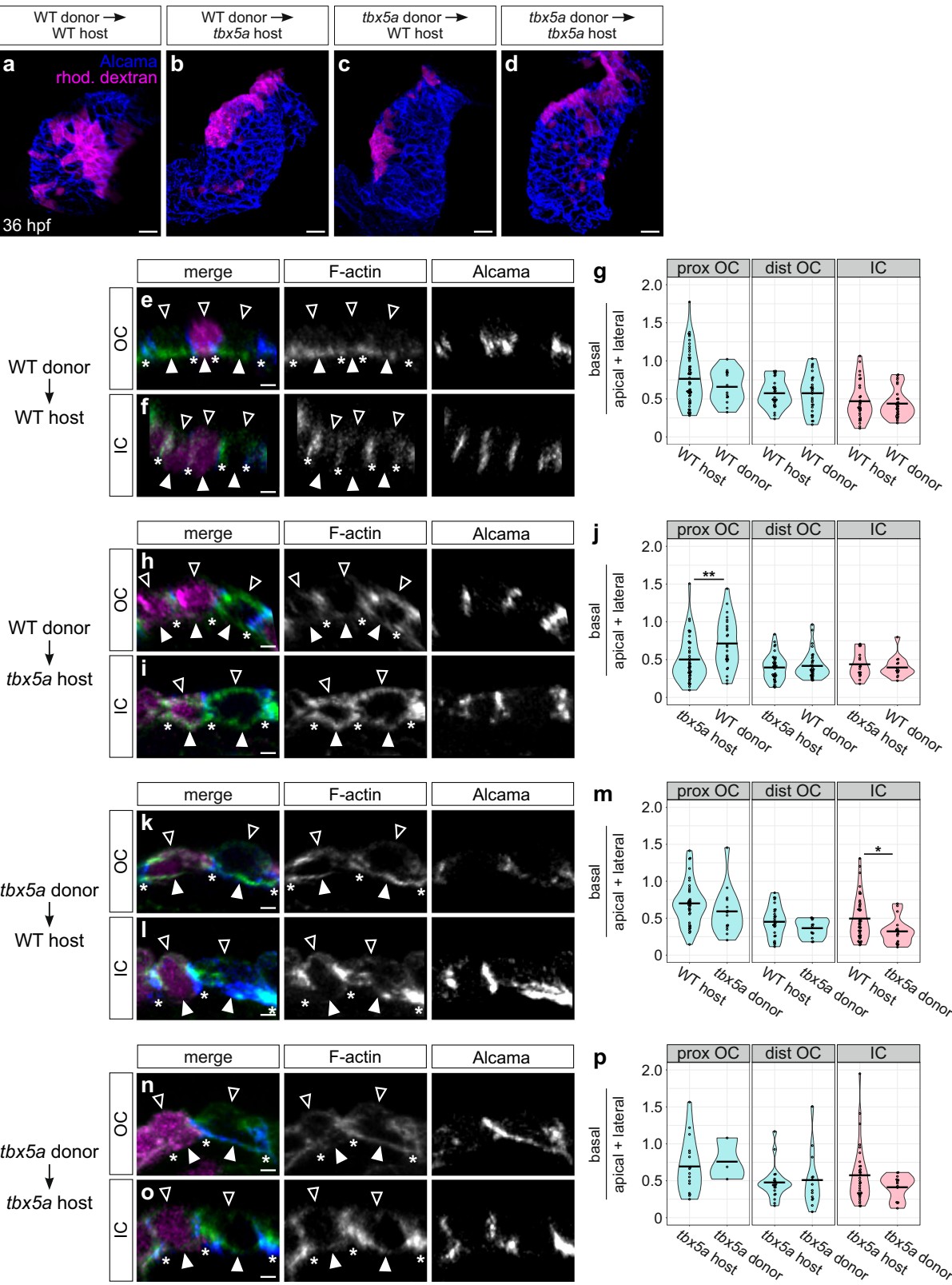

cardiomyocytes, whereas wild-type cardiomyocytes in a wild-type OC send much smaller projections that are usually restricted to cell junctions (Supplementary Figs. 16 and 17). This suggests that wild-type OC cardiomyocytes have a mechanism for sending projections, and that *tbx5a* mutant OC cardiomyocytes (which, at the earlier stage of 36 hpf, have a reduced pool of basal actomyosin (Fig. 7 and Supplementary Fig. 13j)) may have a weakened connection to the underlying ECM. In

addition, we found that cardiomyocytes expressing *DN-myl9* (which also exhibit dysregulated actomyosin localization at 36 hpf (Supplementary Fig. 9)) do not undergo proper planar expansion when surrounded by either wild-type cardiomyocytes (Fig. 3) or other *DN-myl9*-expressing cardiomyocytes (Supplementary Fig. 7). These data suggest that cells with reduced actomyosin activity or disrupted F-actin organization are not capable of the outward pushing behavior that we

**Fig. 8 | *tbx5a* functions in a partially cell-autonomous manner to support subcellular F-actin organization. a–d** 3D reconstructions show examples of mosaic 36 hpf hearts resulting from blastomere transplantation. Immunostaining for Alcama labels the lateral membranes of cardiomyocytes (blue), and donor-derived cells are labeled with rhodamine dextran (magenta). **e, f, h, i, k, l, n, o** Cross-sections through representative cardiomyocytes from the OC (**e, h, k, n**) and IC (**f, i, l, o**) from each of the four transplant scenarios. Immunostaining for Alcama labels lateral membranes of cardiomyocytes (blue) and staining with Phalloidin labels F-actin (green); rhodamine dextran labels donor-derived cardiomyocytes (magenta). Empty arrowheads: apical membranes. Filled arrowheads: basal membranes. Asterisks: basal extreme of lateral membranes. (**g, j, m, p**) Violin plots compare calculated values of (mean basal F-actin/(mean apical F-actin + mean lateral F-actin)) of host-derived cardiomyocytes to those of donor-derived cardiomyocytes for each transplant scenario. Each dot represents an individual cell. Two-sided Wilcoxon test. **j** $p = 0.00524$. **m** $p = 0.0221$. Data gathered from 8 days of transplantation. For WT into WT transplants: host proximal OC ($N = 6$ embryos, $n = 55$ cells); donor proximal OC ($N = 7$ embryos, $n = 14$ cells); host distal OC ($N = 5$ embryos, $n = 30$ cells); donor distal OC ($N = 6$ embryos, $n = 29$ cells); host IC ($N = 4$ embryos, $n = 30$ cells); donor IC ($N = 4$ embryos, $n = 29$ cells). For WT into *tbx5a* transplants: host proximal OC ($N = 5$ embryos, $n = 40$ cells); donor proximal OC ($N = 5$ embryos, $n = 25$ cells); host distal OC ($N = 5$ embryos, $n = 40$ cells); donor distal OC ($N = 5$ embryos, $n = 35$ cells); host IC ($N = 5$ embryos, $n = 18$ cells); donor IC ($N = 5$ embryos, $n = 14$ cells). For *tbx5a* into WT transplants: host proximal OC ($N = 7$ embryos, $n = 38$ cells); donor proximal OC ($N = 5$ embryos, $n = 14$ cells); host distal OC ($N = 6$ embryos, $n = 28$ cells); donor distal OC ($N = 5$ embryos, $n = 13$ cells); host IC ($N = 5$ embryos, $n = 44$ cells); donor IC ($N = 5$ embryos, $n = 18$ cells). For *tbx5a* into *tbx5a* transplants: host proximal OC ($N = 4$ embryos, $n = 16$ cells); donor proximal OC ($N = 2$ embryos, $n = 3$ cells); host distal OC ($N = 4$ embryos, $n = 21$ cells); donor distal OC ($N = 4$ embryos, $n = 14$ cells); host IC ($N = 5$ embryos, $n = 36$ cells); donor IC ($N = 5$ embryos, $n = 14$ cells). Scale bars = 20 μm (**a–d**); 3 μm (**e, f, h, i, k, l, n, o**).

propose enables planar expansion in the OC. Intriguingly, recent work has identified actin-based protrusions that drive the elongation of zebrafish atrial cardiomyocytes as the atrial wall thickens after 72 hpf[77], and we are keen for our future work to explore whether similar protrusions promote planar expansion in the ventricular OC. Another intriguing possibility is that the OC cardiomyocyte projections could help to advance the cellular intercalations involved in the twisting of the heart tube; consistent with this idea, *tbx5a* mutant embryos exhibit reduced cardiomyocyte intercalation and cardiac torsion[9].

Although our model of outward pushing contrasts with the frequent demonstration that actomyosin constricts cell boundaries during morphogenesis, the idea that a growing actin cytoskeleton could force cardiomyocyte membranes outward during curvature formation was originally put forth by Latacha and colleagues nearly two decades ago[37]. Additionally, we know that this effect of the actin cytoskeleton is necessary in other contexts. For example, in the leading edge of migrating cells, actin branching, polymerization, and depolymerization pushes the membrane outward, specifically in the direction of movement[78–80]. These pushing forces are also observed in epithelial contexts: in the *Xenopus* epidermis, cell-autonomous actin dynamics at the apical surface drive outward pushing and thus apical emergence of multiciliated cells[81,82], and, in *Drosophila* follicle cells, actomyosin and its association with the ECM promote expansion of the basal surface, resulting in cell flattening and tissue elongation[83,84]. It would be valuable for future studies to explore potential mechanistic similarities between these contexts and the planar spread of OC cardiomyocytes.

Our data clearly indicate that both extrinsic and intrinsic factors contribute to the regulation of the divergent cytoskeletal landscapes observed in the ventricular curvatures. For the former, we demonstrated that mutants with reduced blood flow through the ventricle not only fail to acquire the expected OC cell morphologies, as previously reported[25], but also exhibit earlier defects in the OC actomyosin network (Fig. 5). Specifically, in *myh6* mutants, both pMyosin and F-actin levels are reduced throughout the ventricular myocardium (Supplementary Fig. 12) and are shifted away from the basal membrane to the lateral and apical membranes in OC cardiomyocytes (Fig. 5). For the latter, we found that *tbx5a* mutants, already appreciated for their underformed curvatures[9,76], also exhibit reduced F-actin levels in the ventricular myocardium (Supplementary Fig. 13), shifted F-actin away from the basal membrane in OC cardiomyocytes (Fig. 7), and defects in the planar expansion of OC cardiomyocytes (Fig. 6). Additionally, within the cardiomyocytes of both *myh6* and *tbx5a* mutants, it appears that G-actin is available at levels similar to that of wild-type (Supplementary Figs. 12 and 13), indicating that actin modifiers fail to organize the monomers into the stereotypical amount and organization. It is interesting to note that reduced blood flow and loss of *tbx5a* function both primarily impacted actomyosin organization in cells of the OC; although both curvatures are exposed to blood flow and *tbx5a* is expressed throughout the ventricular myocardium[76], these factors do not influence the cytoskeleton in IC cardiomyocytes to the same extent that they do in OC cardiomyocytes. This intimates the existence of yet unidentified curvature-specific factors that could help translate extrinsic and intrinsic influences into cytoskeletal changes.

Regarding blood flow, this biomechanical input may be transmitted to the myocardial cytoskeleton through strain on the chamber or through ECM-mediated myocardial-endocardial signaling, but it is unclear what molecules mediate this conversation. It will therefore be beneficial to understand the regional differences in gene expression that exist in the OC and IC, both before and during curvature formation. More specifically, it will be valuable for future work to identify and investigate a roster of actomyosin modifiers whose expression or activity may be regulated, even indirectly, by blood flow or by *tbx5a*. One interesting candidate is *adducin3a* (*add3a*), which is regulated by a heartbeat-dependent microRNA, *miR-143*, and encodes an F-actin capping protein that stabilizes actin polymers[33,85,86]. Overexpression of *add3a* has been shown to prevent planar expansion of OC cells, likely due to the hyperstabilization of the actin cytoskeleton[33], suggesting that regionalized regulation of Add3a activity could contribute to the normal acquisition of OC cell morphology. Finally, it will be interesting to investigate whether any TBX5 targets that may mediate myosin phosphorylation in the mouse second heart field epithelium[58] also function downstream of Tbx5a to modify actomyosin in the OC.

Although actin dynamics have long been implied as a driver of chamber curvature formation, the role of NMII has been debated, with some studies dismissing its role and others supporting it. For example, chick embryos or explanted hearts treated with inhibitors of NMII just after the initiation of curvature formation go on to form relatively normal curvatures[87]. By contrast, explanted zebrafish heart tubes treated with Blebbistatin fail to form chamber curvatures[38]. Of course, these disparate findings could simply signify differences between species. However, our studies offer an alternative explanation in which there are at least two distinct phases of the influence of actomyosin on the regulation of cardiomyocyte shape. Our treatments with Blebbistatin or Latrunculin B (Supplementary Fig. 6) imply that, early in curvature formation (~24–36 hpf), both actin polymerization and NMII activity are required to allow the subsequent planar expansion of OC cardiomyocytes. In contrast, later in curvature formation (~36–48 hpf), actin polymerization does not seem to impact OC cell planar expansion, and NMII even plays the opposing role of restricting planar spread, presumably due to its canonical contractile properties (Fig. 9). These findings, together with the impact of *DN-myl9* expression (and a corresponding reduction of pMyosin (Supplementary Fig. 8)) on both total F-actin (Supplementary Fig. 8) and subcellular localization (Supplementary Fig. 9), support a model wherein the early F-actin dynamics that are crucial for subsequent OC cell shape change may be regulated by NMII. The influence of NMII on F-actin dynamics is a commonly

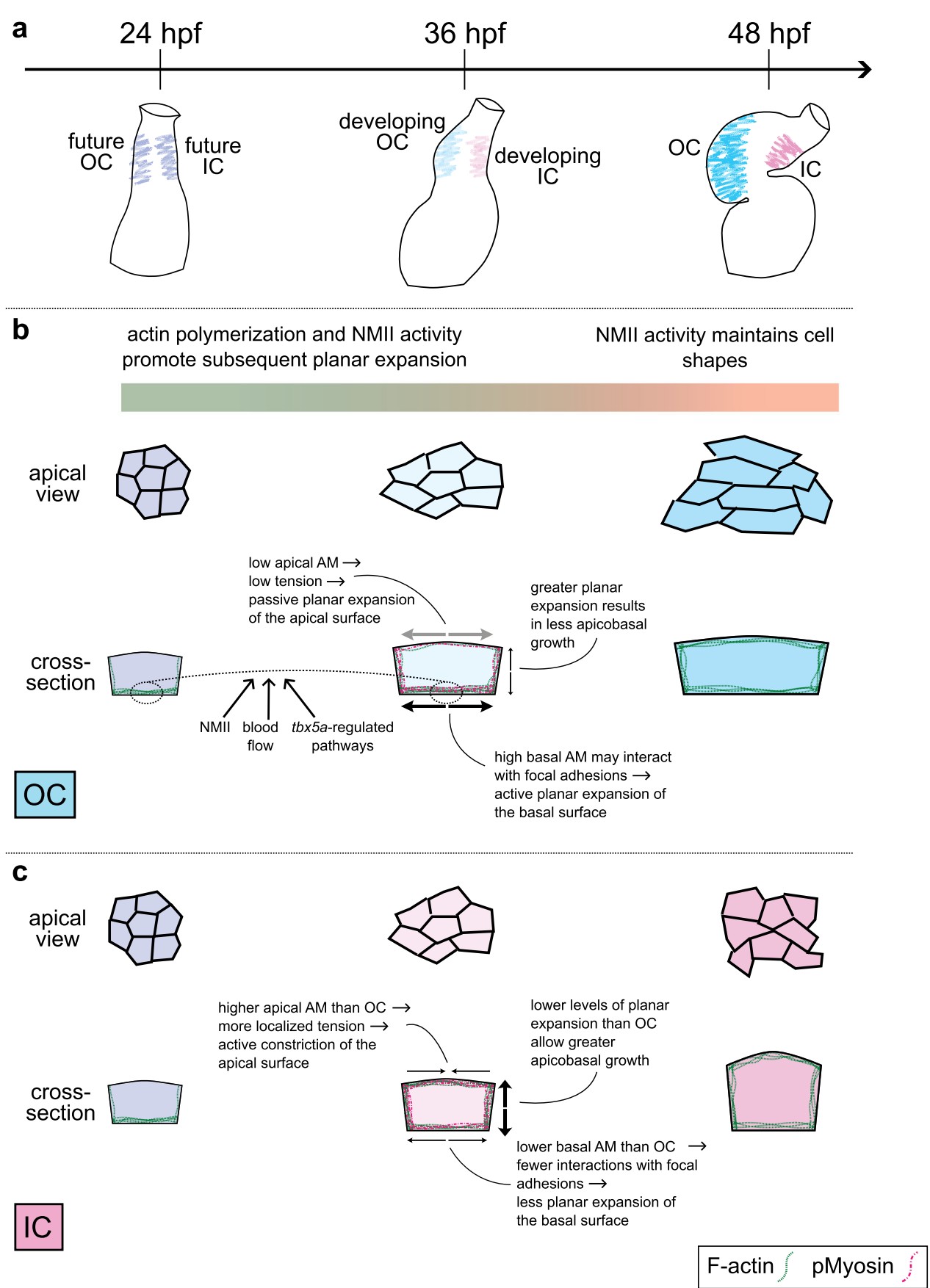

**Fig. 9 | Changes to the actin cytoskeleton and cardiomyocyte shape during curvature formation. a** During the second day of zebrafish development, the linear heart tube transforms into a two-chambered heart. Once similarly contoured regions (light purple) bend and stretch to become the convex OC (blue) and the concave IC (pink). **b, c** Schematic portrays select cellular and subcellular traits and events that accompany curvature formation; see Discussion for more detail. At the linear heart tube stage, cardiomyocytes in the future OC and IC (particularly those positioned more proximally, as shown here; light purple) present similar morphologies and F-actin organization, with most F-actin restricted to the basal surface. Over the next 12 h, OC (**b**, light blue) and IC (**c**, light pink) cardiomyocytes have begun to diverge slightly in shape, and F-actin organization has also diverged. In the OC, there remains a large pool of basal F-actin, whereas in the IC, a lower proportion of F-actin resides at the basal surface and there exists a larger pool of apical and lateral F-actin. This difference is facilitated by NMII activity, as well as by blood flow through the ventricle and by *tbx5a*-regulated pathways. We propose

that the basal enrichment of F-actin in OC cardiomyocytes (**b**) allows for interaction with focal adhesions to enact outward pushing of the basal surface (thick black arrows). Simultaneously, maintaining a low amount of actomyosin at the apical surface may allow this surface to passively expand along with the actively spreading basal surface (gray arrows). In the IC (**c**), we speculate that lower levels of basal F-actin might result in weakened interaction with focal adhesions and less outward pushing (thin black arrows), while more actomyosin at the apical surface may increase tension and prevent passive spreading (inward-facing thin black arrows). As a consequence of limited planar expansion, the increasing volume of IC cardiomyocytes instead leads to expansion along the apicobasal axis (thick black arrows). By 48 hpf, OC (darker blue) and IC (darker pink) cardiomyocytes have acquired strikingly different cell morphologies. In contrast, the actin cytoskeleton has converged into similar arrangements in OC and IC cardiomyocytes, with F-actin distributed fairly uniformly around all membranes.

reported phenomenon[66–74], and we therefore recommend special attention to the timing of any such actomyosin manipulations in studies of curvature formation. Indeed, the perduring expression of *DN-myl9* throughout curvature formation in our experiments may cloud our interpretations, as its dominant negative effect during later curvature formation could suppress the effects of earlier NMII activity (and vice versa). In future studies, having both spatial and temporal control over actomyosin cytoskeleton modification will help to test these ideas more directly.

Cardiac looping involves the coordination of several morphogenetic events in both the ventricle and the atrium, as well as in structures like the OFT and AVC. However, the degree of interdependence of events in these different domains remains unclear. Ex vivo studies in zebrafish suggest that accretion of second heart field cells to the outflow and inflow tract regions are not required for the repositioning of chambers, torsion, or formation of the curvatures[9,38], implying that these processes exhibit some level of independence. In contrast, modulation of Bmp signaling in the AVC and IC leads to reduced chamber displacement, including a fairly unkinked IC[88], highlighting a link between AVC and IC morphogenesis. Our studies show that it is primarily the OC that is influenced by blood flow and *tbx5a* (Fig. 4–7 and Supplementary Fig. 11) and that the cell-autonomous effects of *tbx5a* and of *DN-myl9* expression are only observed in the OC (Figs. 3, 8 and Supplementary Fig. 15), suggesting that the pathways studied here are mostly relevant to OC development. Taking these data together, we speculate that the shape of the IC is largely coupled to the morphogenesis of the AVC, whereas OC dynamics and shape are at least partially driven by the cell-intrinsic, actomyosin-based mechanisms presented here. However, we cannot rule out the possibility of interplay between OC and IC morphogenesis or the potential interaction of OC formation with other elements of looping, such as torsion and chamber displacement.

Finally, we note that, although our data highlight distinct actomyosin landscapes in the proximal and distal regions of the OC at 36 hpf, cells in the proximal and distal regions of the OC at 48 hpf attain quite similar morphological attributes. In addition, disrupting actomyosin function affects cardiomyocyte shape change in both proximal and distal OC regions. We therefore speculate that additional mechanisms may drive the cell and tissue shape changes in these regions of the OC. One possible player may be the underlying ECM, known for its multiple roles in cardiac morphogenesis[89]. The cardiac jelly has the potential to impact curvature formation at several levels, including at the level of actomyosin structure and activity[90], at the level of cell spreading[91], and at the level of tissue curvature[92,93]. These feats may be accomplished by providing the physical cues to stimulate mechanotransduction pathways or by acting as a signaling center[94]. Indeed, various ECM components are regionalized within the developing wild-type ventricle[95,96], suggesting potential differences in the role of the ECM role between, and perhaps within, the OC and IC.

Future work clarifying how ECM dynamics control the regionalized cytoskeletal and cell morphology changes observed here will be vital to understanding chamber curvature formation as a whole.

Altogether, our work supports the notions that intrinsic regulators and extrinsic inputs converge to control the actomyosin cytoskeleton in the OC, that these events are crucial for changes to cell morphology, and that these changes ultimately influence the proper sculpting of the ventricle. These findings offer a mechanistic framework both for the specific circumstance of cardiac chamber emergence and for tissue curvature formation in general. In addition, they provide a foundation for further articulation of the pathways that create the distinctions between the OC and IC, which could ultimately shed light on the origins of certain types of congenital defects in chamber morphology and could also enhance efforts to engineer ventricular tissue in vitro.

## Methods

### Zebrafish

The following transgenic lines were used in this study: *Tg(myl7:WT-myl9-mScarlet)*[bnsS24Tg32], *Tg(myl7:DN-myl9-eGFP)*[bns333Tg32], *Tg(myl7:eGFP-Hsa.HRAS)*[s883Tg97], and *Tg(myl7:mKate-CAAX)*[sd11Tg98]. We also used lines carrying the *tbx5a*[m21] and *myh6*[m58] mutations. Heterozygous carriers of mutations were identified by PCR amplification and subsequent restriction enzyme digest, as previously described for *tbx5a*[m21][30] and *myh6*[m58][98]. Homozygous mutant embryos were identified by previously characterized morphological phenotypes[75,76]. All work presented here followed protocols (#S09125) approved by the Institutional Animal Care and Use Committee at the University of California, San Diego.

### Pharmacological treatments

Embryos were dechorionated and placed in 6-well culture dishes and exposed to either 1.25% DMSO, 1.25% DMSO + 100 ng/mL Latrunculin B (Millipore Sigma, L5228), or 1.25% DMSO + 30 μM Blebbistatin (Millipore Sigma, B0560) in E3 medium. DMSO controls were kept in solution between 24 and 48 hpf. Latrunculin B and Blebbistatin-treated embryos were either kept in drug-containing media between 24 and 36 hpf and switched to 1.25% DMSO between 36 and 48 hpf, or vice versa. At 48 hpf, all embryos were fixed and processed as described below.

### Immunofluorescence and phalloidin, and DNase I staining

Embryos were dechorionated, fixed in 1% methanol-free formaldehyde (ThermoFisher Scientific, 28908) for 1 h and 10 min with gentle rocking, and then rinsed three times in PBT (1X PBS containing 0.1% Tween 20 (Sigma-Aldrich, P9416)). Embryos were rocked rapidly in a 0.2% saponin solution (Sigma-Aldrich, S4521) in PBT containing 0.5% Triton X-100 (G-Biosciences, 786513) until yolks were dissolved. This took 20–35 min and was monitored under a dissecting microscope. Embryos were rinsed three times in PBT, placed in 4% formaldehyde, and incubated at 4 °C overnight. On the following day, to improve the

antibody signal-to-noise ratio, embryos were rinsed three times in PBT, placed in prechilled 100% acetone, and incubated at −20 °C for 8 min. Acetone was replaced with PBT + 0.5% Triton X-100 for one hour at room temperature. To ensure reagent access to the heart, forceps were used to open the pericardial cavity. Embryos were then blocked with 2 mg/ml bovine serum albumin (Sigma-Aldrich, A9647) and 10% goat serum in PBT for at least 1 h at room temperature, then incubated with primary antibody in block overnight at 4 °C. Embryos were then washed extensively with PBT, blocked as described for primary antibody incubation, and incubated with the secondary antibodies in blocking solution overnight at 4 °C. Embryos were then washed extensively with PBT.

The following primary antibodies were used at the specified dilutions: mouse anti-Alcama (Developmental Studies Hybridoma Bank, Zn-8 supernatant, 1:50); rabbit anti-Cdh2 (GeneTex, GTX125885, 1:200); rabbit anti-phospho-Myosin (Abcam, ab2480, 1:100); rabbit (Life Technologies, A11122, 1:500) or chicken (Life Technologies, A10262, 1:1000) anti-GFP; rabbit anti-dsRed (also detects mScarlet; Clontech, 632496, 1:1000); rabbit anti-TagRFP (also detects mKate; Evrogen, AB233, 1:500); mouse anti-myosin heavy chain (Developmental Studies Hybridoma Bank, MF20 supernatant, 1:50); and mouse anti-Myh6 (Developmental Studies Hybridoma Bank, S46 supernatant, 1:50).

All secondary antibodies were produced in goat by Thermo Fisher Scientific and used at a dilution of 1:300. Secondary antibodies used were: anti-mouse-AlexaFluor 488 (A11001); anti-mouse-AlexaFluor 568 (A11031); anti-mouse-AlexaFluor 647 (A21236); anti-rabbit AlexaFluor 488 (A11008); anti-rabbit AlexaFluor 568 (A11011); anti-rabbit Alexa-Fluor 647 (A21245); and anti-chicken AlexaFluor 488 (A11039). In some cases, Phalloidin-AlexaFluor 488 or Rhodamine-Phalloidin (Life Technologies, A12379 or R415; each reconstituted to 200 U/mL in MeOH) was added to the secondary antibody solution at a dilution of 1:30.

For Supplementary Figs. 8, 12, and 13, embryos were stained with DNase I-AlexaFluor 594 (Thermo Fisher D12372; reconstituted to 5 mg/mL in phosphate-buffered saline, pH 7.4 (PBS) + 50% v/v glycerol), diluted 1:500 to detect G-actin[63,64], and Phalloidin-AlexaFluor 647 (ThermoFisher A22287; reconstituted to 66 μM in DMSO), diluted 1:300 to detect F-actin. Staining was performed similarly to immunostaining, but without the addition of Triton X-100 to any solutions to prevent the extraction of G-actin[65] and without a blocking step. Although the Phalloidin signal was robust in both myocardium and endocardium, the DNase I signal was detected at substantially higher levels in the myocardium when compared with the endocardium, supporting the specificity of this reagent.

## Fluorescent in situ hybridization

Fluorescent in situ hybridization was performed as previously described[99], but embryos were fixed and deyolked as for immuno-fluorescence before MeOH dehydration and were incubated with Proteinase K (50 μg/mL) in PBT for 2 min at room temperature prior to hybridization with the probe. Probe for *nppa* (ZDB-GENE-030131-95) was produced as previously described[75]. Probe for *mb* (ZDB-GENE-040426-1430) was produced by PCR amplification of 48 hpf embryonic cDNA followed by T7 in vitro transcription. The forward primer used was 5′-GGACAAACACCGCGACAGAC-3′ and the reverse primer used was: 5′-TAATACGACTCACTATAGGGCCTGAGACCCTAACGA ACCATTAT-3′; underlined portion is the T7 binding site.

## Transplantation

Blastomere transplantation was performed similarly to a previous description[25]. Briefly, donor embryos were injected with 5% lysine-fixable rhodamine dextran (ThermoFisher Scientific, D1817) in 0.2 M KCl at the one-cell stage. Approximately 10–30 blastomeres were taken from dechorionated 3–4 hpf donor embryos and placed into the margin of similarly staged host embryos. Host embryos were raised in E3 medium containing penicillin-streptomycin (Millipore Sigma, P4458; diluted 1:50), in agarose-coated wells. Host embryos were allowed to develop until 36 or 48 hpf when they were screened for the presence of donor-derived cells in the myocardium, which could be identified by fluorescence under a Zeiss Axiozoom microscope. Hosts containing donor-derived cardiomyocytes were fixed and processed for immunostaining as described above. In order to be included for analysis, cardiomyocytes needed to meet specific criteria. For donor-derived cells, we analyzed any cell with at least half of its body within the OC or IC boundary. For host-derived cells, we analyzed any cell (1) in direct contact with a donor-derived cell or one cell-distance from a donor-derived cell and (2) with at least half of its body within the OC or IC boundary.

For *Tg(myl7:DN-myl9-eGFP)* transplants, transgenic animals were crossed to wild-type animals, and the resulting embryos served as donors; wild-type embryos served as hosts. For the *tbx5a* transplantation experiments, animals heterozygous for the *tbx5a* mutation were crossed to obtain both donor and host embryos. After transplantations were complete, donor embryos were genotyped by PCR and restriction enzyme digest for the *tbx5a* mutation, as described above. Genotypes of host embryos were determined based on the presence or absence of pectoral fin buds at 36 hpf or pectoral fins at 48 hpf[76]. While it is possible that a large donor-derived clone could affect the development of one pectoral fin bud, it is unlikely, given the number of donor cells transplanted, that this would impact both pectoral fin fields in the same host. Both sides were therefore assessed for the presence of pectoral fin buds/fins, and only those host embryos that had matching sides were analyzed.

## Transgene plasmid injection

To obtain the mosaic expression of *Tg(myl7:DN-myl9-mScarlet)*[32] shown in Supplementary Fig. 10, a mixture of transgene plasmid (10–20 pg per embryo) and *tol2* mRNA (20–60 pg per embryo) was injected into the yolk of wild-type embryos at the one-cell stage. Embryos were screened for mScarlet fluorescence at 48 hpf, then were fixed and immunostained for Alcama as described above.

## Image acquisition

After all immunostaining and fluorescent in situ hybridization procedures, hearts were dissected from embryos under a Zeiss AxioZoom microscope. To keep organ morphology as intact as possible, hearts were simply mounted in a droplet of PBT on a cover slip and immediately imaged using a Leica SP8 laser-scanning confocal microscope and a 25X water-immersion objective. Hearts were positioned laterally, with one side of the ventricle flat against the coverslip. In all experiments where signal intensity was measured, laser settings were kept consistent between samples. Z-stacks were captured with a slice thickness of 0.3 μm.

## Image analysis

All image files captured by the Leica SP8 confocal were rendered as 3D reconstructions in Imaris v11.0.0 (Bitplane). Often, secondary analysis was completed in FIJI v2.16.0 (ImageJ).

**Defining the ventricular OC and IC.** To develop a method to standardize the boundaries of the OC and IC between 48 hpf samples (Supplementary Fig. 1e), we first assessed the expression domain of *nppa* in 12 wild-type hearts at 48 hpf (Supplementary Fig. 1a, b). By using the following measurements, we found that we could capture, on average, the highest *nppa* signal in the resulting "OC" and, on average, the lowest *nppa* signal in the resulting "IC". These territories also corresponded well to the regions with the lowest (OC) and highest (IC) *mb* expression (Supplementary Fig. 1c, d). These measurements were then applied to 48 hpf hearts throughout this study.

The following abbreviations are used in our description of this measurement method: PB$_{OC}$: OC proximal boundary; DB$_{OC}$: OC distal boundary; DB$_{OFT(OC)}$: distal boundary of the OC side of the OFT; PB$_{IC}$: IC proximal boundary; DB$_{IC}$: IC distal boundary; DB$_{OFT(IC)}$: distal boundary of the IC side of the OFT.

To delineate the OC in Imaris (Supplementary Fig. 1e), the arc of the ventricular OC was measured beginning one cell diameter from the AVC (the PB$_{OC}$) to the DB$_{OFT(OC)}$. This arc is represented by a blue dotted line in Supplementary Fig. 1e. Starting from the PB$_{OC}$, the DB$_{OC}$ was marked at 2/3 of the distance from the PB$_{OC}$ to the DB$_{OFT(OC)}$. An *Oblique Slicer* was placed between the PB$_{OC}$ and DB$_{OC}$, ensuring that it was perpendicular to the arc of the OC. The position of the *Oblique Splicer* is represented by a solid blue line in Supplementary Fig. 1e. The portion of tissue bounded by the solid blue line was considered the OC.

To delineate the IC in Imaris (Supplementary Fig. 1e), the arc of the ventricular IC was measured from the AVC (the PB$_{IC}$) to the DB$_{OFT(IC)}$. This arc is represented by a dotted pink line in Supplementary Fig. 1e. Point I (red "I" in Supplementary Fig. 1e) was marked at 1/2 of the distance between PB$_{OC}$ and DB$_{OC}$. Point II (red "II" in Supplementary Fig. 1e) was marked at 1/3 of the distance between PB$_{IC}$ and Point I. The proximal border of the IC was considered to run from the PB$_{IC}$ to Point II. This border is represented by a solid pink line in Supplementary Fig. 1e. Starting from the PB$_{IC}$, the DB$_{IC}$ was marked at 1/2 of the distance from the PB$_{IC}$ to the DB$_{OFT(IC)}$. Point III (red "III" in Supplementary Fig. 1e) was marked halfway between the DB$_{IC}$ and DB$_{OC}$. The distal border of the IC was considered to run from the DB$_{IC}$ to Point III. This border is also represented by a solid pink line in Supplementary Fig. 1e. Finally, an *Oblique Slicer* was placed between Points II and III, ensuring that it was perpendicular to the arc of the IC. The position of this *Oblique Splicer* is represented by a solid pink line in Supplementary Fig. 1e. The portion of tissue bounded by the solid pink lines was considered the IC.

To account for the reduced contours of the OC and IC in hearts at 36–37 hpf (Supplementary Fig. 1h), slight adjustments were made in our measurement method. Specifically, the arc of the ventricular OC was measured beginning one cell diameter from the AVC (the PB$_{OC}$) to the DB$_{OFT(OC)}$. This arc is represented by a dotted blue line in Supplementary Fig. 1h. Starting from the PB$_{OC}$, the DB$_{OC}$ was marked at 2/3 of the distance from the PB$_{OC}$ to the DB$_{OFT(OC)}$. The arc of the ventricular IC was measured from the AVC (the PB$_{OC}$) to the DB$_{OFT(IC)}$. This arc is represented by a dotted pink line in Supplementary Fig. 1h. A line was drawn between the PB$_{OC}$ and the PB$_{IC}$, as well as between the DB$_{OC}$ and the DB$_{IC}$; these are represented by dotted purple lines in Supplementary Fig. 1h. Point I (red "I" in Supplementary Fig. 1h) was marked at 1/3 of the distance between PB$_{OC}$ and PB$_{IC}$, and Point II (red "II" in Supplementary Fig. 1h) was marked at 2/3 of the distance between PB$_{OC}$ and PB$_{IC}$. Point III (red "III" in Supplementary Fig. 1h) was marked at 1/3 of the distance between DB$_{OC}$ and DB$_{IC}$, and Point IV (red "IV" in Supplementary Fig. 1h) was marked at 2/3 of the distance between DB$_{OC}$ and DB$_{IC}$. The medial border of the OC was considered to run from Point I to Point III (solid blue line in Supplementary Fig. 1h), and the medial border of the IC was considered to run from Point II to Point IV (solid pink line in Supplementary Fig. 1h). As for 48 hpf hearts, *Oblique Slicers* were used in accordance with the medial border measurements in Supplementary Fig. 1h to bound the OC and IC.

**Labeling lateral cell membranes.** When the labeling of lateral membranes was necessary, we utilized one of four reagents: *Tg(myl7:mKate-CAAX)*, *Tg(myl7:Ras-eGFP)*, anti-Cdh2 antibody, or anti-Alcama antibody. The first two reagents label all membranes of the cardiomyocyte, whereas the latter two are cell adhesion molecules and are therefore expected to label only the lateral membranes. Given the possibility that Cdh2 and Alcama could be polarized along the apicobasal axis, we tested the extent to which these signals overlapped with *Tg(myl7:Ras-*

*eGFP)* signal (Supplementary Fig. 2). The consistent overlap of Cdh2 and Alcama signal with lateral Ras-eGFP signal (Supplementary Fig. 2[100];) led us to consider both Cdh2 and Alcama as sufficient representations of the entire lateral membrane, and we therefore used antibodies against either, depending on the specific requirements of an experiment.

**Cardiomyocyte morphometrics.** For each cardiomyocyte in the OC and IC, Imaris was used to capture *Snapshots* of the necessary views of the cell, FIJI was used to take measurements, and Excel was used to compile these measurements into a.csv and to calculate apicobasal length and volume.

In Imaris: To capture the *en face* view to obtain apical surface area and circularity, a single *Snapshot* of the apical surface of the cardiomyocyte was taken, ensuring that the surface was flat and parallel to the screen. If a cardiomyocyte was large enough that it curved in the Z-plane, multiple *Snapshots* were captured. To do this, an *Oblique Slicer* (or more, if necessary) was placed where the greatest curvature existed, perpendicular to the long axis of the cell. Then two (or more, if necessary) *Snapshots* of the cell were taken–the first on one side of the *Oblique Slicer*, and a second on the other side of the *Oblique Slicer*, ensuring that each "side" of the cell was flat and parallel to the screen. To capture the cross-sectional view to obtain apicobasal length and volume measurements, two *Oblique Slicers* were placed while the cell was still in an *en face* view. These *Oblique Slicers* were perpendicular to the apical surface of the cell and approximately bisected the cell (one bisected the major axis and the other bisected the minor axis). The *Volume* of the 3D reconstruction was clicked *Off*, and the image was reoriented so that the "faces" of the *Oblique Slicers* were visible, showing cross-sections of the cell. The intersection of the two *Oblique Slicers* marked the current cell of interest. Two *Snapshots* of the cell were taken, one from each *Oblique Slicer*, ensuring that the *Slicers* were flat and parallel to the screen.

In FIJI: The scale was first set based on the scale bar provided by Imaris. To measure apical surface area and circularity, the *Freehand Selection* tool was used to trace the lateral membrane of the cardiomyocyte. If a cardiomyocyte was curved in the Z-plane and required two (or more, if necessary) *Snapshots*, the lateral membrane of one "side" of the cell was traced, stopping where the cell meets the *Oblique Slicer*. The selection was copied and pasted onto the *Snapshot* of the other "side", the "sides" were positioned so that they joined together, and the border of the entire cell was traced. The *Measure* function was used to obtain *Area* and *Circularity* values. For each of the two cross-sectional *Snapshots* taken to measure apicobasal length, the basal width (the distance between the basal tips of the lateral membranes), apical width (the distance between the apical tips of the lateral membranes), and length of each lateral membrane (from basal tip to apical tip) were measured using the *Measure* function.

In Excel: To obtain apicobasal length, the average (mean) of the lengths of the four lateral membranes was calculated. To obtain volume, the average (mean) of the two basal width values and the two apical width values was calculated; then, the mean basal width, the mean apical width, and the mean apicobasal length were multiplied.

**Subcellular actomyosin localization.** Prior to image analysis, F-actin and pMyosin (where appropriate) signal in each ventricle was assessed in Imaris. Briefly, a *Mask* of the ventricular myocardium was generated, the *Mean* signal intensity for each channel was located in the *Statistics* tab, and these values were tested for outliers in R[101]. Outliers were removed from further analysis.

For each cardiomyocyte in the OC and IC, Imaris was used to capture *Snapshots* of the necessary views of the cell, FIJI was used to take signal intensity measurements, and Excel was used to compile these measurements into a.csv and to calculate mean signal intensities.

In Imaris: Two cross-sectional views (Fig. 2a, b) were captured as described above in the "*Cardiomyocyte morphometrics*" section, with the exception that a separate *Snapshot* was taken for each channel (for the membrane marker, for F-actin, and for pMyosin).

In FIJI: All images were converted to 8-bit and the channels for each *Snapshot* were *Merged*. Using the membrane marker channel as a guide, the *Freehand Line* tool, set to a width of ~1 μm, was used to trace just inside the membrane. The *Mean* signal intensities of the F-actin signal and the pMyosin signal were measured along each membrane using the *Measure* function, toggling between channels as needed. Four *Mean* measurements were taken per *Snapshot*: basal, apical, and two lateral. To prevent the inclusion of signal from the lateral membranes into the basal/apical datasets and vice versa, a ~1 μm segment in each "corner" (where the lateral and basal/apical membranes intersect) was avoided when performing the initial traces.

In Excel: For F-actin and for pMyosin (when present) in each cell, the averages (means) of the two basal signal intensities, of the two apical signal intensities, and of the four lateral signal intensities were calculated.

**Total pMyosin, F-actin, and G-actin in the ventricular myocardium.** All analysis was completed in Imaris v11.0.0. A *Surface* of the myocardium was generated using a *Surfaces Detail* of 0.9 μm and *Machine Learning* segmentation to include the ventricular myocardium and to exclude the endocardium as well as the space around the myocardium. The *Surface* was *Cut* at the AVC/ventricle boundary and at the ventricle/OFT boundary. The *Mean* signal intensity of each channel was located in the *Statistics* tab. Finally, each experimental replicate was normalized to a scale of 1–10 relative fluorescence units (rfu) using min-max normalization. This transformation allows for comparison of experimental replicates performed on different days while preserving the relationship between datapoints.

**Comparing the morphologies of OC and IC cardiomyocytes with different genotypes**
In Supplementary Fig. 11, we sought to compare cardiomyocyte morphologies of *myh6* and *tbx5a* mutants to wild-type, all on one scale. To accomplish this comparison, we normalized the experimental mutant data using the formula:

$$\text{mean}_{\text{norm.exp}} = \text{mean}_{\text{ref}} + (\text{SD}_{\text{ref}} * \text{mean}(Z\text{-score}_{\text{exp}})),$$

where the $\text{mean}_{\text{ref}}$ is the mean of the 48 hpf wild-type reference dataset presented in Fig. 1, the $\text{SD}_{\text{ref}}$ is the standard deviation of the 48 hpf wild-type reference dataset presented in Fig. 1, and the $\text{mean}(Z\text{-score}_{\text{exp}})$ is the mean $Z$-score of the experimental group (from the data presented in either Figs. 4 or 6). For example, the mean apical surface area for the 48 hpf wild-type OC reference dataset in Fig. 1 is 142 μm², and the standard deviation for this dataset is 43 μm². In Fig. 6, the mean apical surface area for the 48 hpf wild-type sibling OC dataset is 147 μm², the mean apical surface area for the *tbx5a* mutant OC dataset is 108 μm², and the mean $Z$-score for this experimental group is −0.67 (i.e., 0.67 standard deviations below the control mean). Therefore, for the apical surface area of *tbx5a* mutant OC cardiomyocytes,

$$\text{mean}_{\text{norm.exp}} = 142\,\mu m^2 + (43\,\mu m^2 \times -0.67) = 113\,\mu m^2$$

$\text{mean}_{\text{norm.exp}}$ values were calculated in this way for apical surface area and apicobasal length for each dataset (*myh6* OC, *myh6* IC, *tbx5a* OC, and *tbx5a* IC) and plotted alongside the raw mean apical surface area and apicobasal length for the 37 hpf and 48 hpf wild-type OC and IC datasets from Fig. 1.

**Statistical analysis, plotting, and figure design**
Statistics were performed in R (v4.4.1)[101] and RStudio (v2025.09.1 + 401)[102]. For all comparisons, statistical significance was determined by the Wilcoxon test, which does not assume normal data distribution. Statistically significant p-values are indicated in graphs by asterisks as follows: * denotes $p < 0.05$; ** denotes $p < 0.01$; *** denotes $p < 0.001$. The exact number of embryos and/or cells examined for each experiment is reported in the corresponding figure legend.

The following R packages were used to restructure, statistically analyze, and plot the quantitative data captured from micrographs: dplyr[103], ggbeeswarm[104], ggplot2[105], ggpmisc[106], ggpubr[107], ggtext[108], Lattice[109], plyr[110], Tidyverse[111], and ggnewscale[112]. In Figs. 1 and 4−7 and Supplementary Figs. 5−7 and 9, SuperPlots[113] were employed to portray variation between samples within the same experimental group. Figures were produced in Adobe InDesign v.18.5.2.107 and Inkscape v1.3.2.

### Reporting summary
Further information on research design is available in the Nature Portfolio Reporting Summary linked to this article.

## Data availability
The data generated in this study are provided in the Source Data file. Source data are provided with this paper.

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

## Acknowledgements

We thank members of the Yelon laboratory for valuable discussions, A. Negrete and M. Ayan for help with genotyping, and A. Yarbrough and the UCSD Animal Care Program for zebrafish care. Funding was provided by the following sources: American Heart Association grant 23TPA1072669 (D.Y.), Saving tiny Hearts Society (D.Y.), National Institutes of Health fellowship F32 HL147435 (D.M.L.), American Heart Association fellowship 20POST35110077 (D.M.L.), Max Planck Society (D.Y.R.S.), EMBO fellowship LTF1569 (R.P.), Alexander von Humboldt Foundation fellowship (R.P.), Cardio-Pulmonary Institute grant EXC 2026 project ID 390649896 (R.P.).

## Author contributions

Conceptualization: D.M.L. and D.Y. Data curation: D.M.L. and D.Y. Formal analysis: D.M.L., G.B.A. and D.Y. Methodology: D.M.L., G.B.A., and D.Y. Investigation: D.M.L., G.B.A., and D.Y. Resources: D.M.L., R.P., and D.Y.R.S. Visualization and validation: D.M.L. and D.Y. Supervision: D.Y.R.S. and D.Y. Writing—original draft: D.M.L. Writing—review & editing: D.M.L., G.B.A., R.P., D.Y.R.S., and D.Y.

## Competing interests

The authors declare no competing interests.
