## [Transparent Peer Review file · Nature Communications]

Regionalized regulation of actomyosin organization influences cardiomyocyte cell shape changes during chamber curvature formation

Corresponding Author: Dr Deborah Yelon

Version 0:

Reviewer comments:

Reviewer #1

(Remarks to the Author)

In this study, Leeberg et al investigate the influence of the actomyosin network on cardiomyocyte cell shape during curvature morphogenesis, with specific emphasis on apicobasal organisation of the cytoskeleton. Using cardiomyocyte morphometric analysis at different times in cardiac development, the authors describe that OC and IC cells diverge in their morphology and show F-actin and p-Myosin organisation precedes cell shape changes, presumably driving ventricular wall curvature. Whilst this is perhaps unsurprising, it is novel and has (to our knowledge) not been demonstrated in this process before. Using pharmacological treatments, transgenics, and mutant analyses combined with cell transplant assays, they investigate the actomyosin network during this process and how it is regulated by both intrinsic and extrinsic factors. The experiments are elegantly performed and analyses carefully conducted, with the interpretation of the data mostly appropriate.

There are two major concerns with the work that require addressing:

1. The work using the myl9 transgenics (DN and CA) are not convincing and at times contradictory. There are several aspects to this:
 - a. The generation of stable DN and CA myl9 transgenics is surprising. If these are major modifiers of the actomyosin network, it's surprising they are viable (for e.g., sustained blebbistatin exposure would likely be lethal). Is there a phenotype in the adult fish? Are they healthy and 100% viable? Is there a difference in the curvature of the heart? If not, it could be argued that these are not the correct model to address central questions of the manuscript.
 - b. DN-myl9 and CA-myl9 are used to investigate the role of NMII in OC and IC cardiomyocyte morphologies. DN-myl9 weakens the actomyosin machinery while CA-myl9 strengthens it. Yet, the use of both transgenes results in similar phenotypes such as decreased apical surface area and F-actin levels. Could the authors comment on this?
 - c. What's the explanation for the reduced fluorescence of F-actin? Could it be squelching or a similar effect that is different from intended?
 - d. What are the expression levels of the transgenics like? Are they weaker than the mosaic or injected F0? Could there be some silencing/inhibition of the transgene occurring that might account for the subtlety of the phenotypes?
 - e. Some effects in the mosaic analysis are opposite to that observed in the transgenic. For e.g. the circularity of stable CA myl9 is decreased in the OC and yet increased in the mosaic analysis. Could the authors comment on this?
2. Whilst this is an elegant study, it is brief, which limits the contribution. Could the authors take the work a little further? For example, is there any information about what Tbx5a might be regulating to control actomyosin organisation? Are there any Formins, Rho GTPases, etc that are known to be downstream of Tbx5? Alternatively, the observation of the basal projections is intriguing and there is a lot of speculation in the Discussion about how this might be regulated. Could this be tested to some extent?

Minor comments:

- The authors show that there are cell shape differences between the OC and the IC. This appears to be driven by the organisation of the actomyosin network. While manipulation of the actomyosin network (by DN-myl9, CA-myl9, blood flow,

tbx5a mutants etc.) results in reduced planar spread in OC cells, localisation of F-actin and in some cases pMyo in the proximal OC is the most affected. Looking more closely in WT hearts, the expression pattern of F-actin and pMyo (fig S3 and figS4D) in the distal OC appears more similar to the IC. This begs the question – are there cell shape differences between the proximal and distal cells of the OC? I.e. is the proximal OC more squamous than the distal OC? Additionally, in treated samples are proximal OC cell shape more disrupted?

- The authors posit that changes in cell shape accompany curvature formation. In several assays conducted by the authors cell shape especially in the OC is affected but its effect on characteristic chamber curvatures is not measured. Could this be addressed?

- Disruption of NMII, blood flow and tbx5a function result in reduced F-actin levels. Is this due to a reduction of actin expression or is actin polymerisation affected? An IF for G-actin or a western blot for actin may aid in answering this.

- Apicobasal length is measured using confocal images. In some instances, it appears as though the images have been acquired with in the X-Y plane. At other times, apicobasal axis appears to be acquired in the Z-axis. Given that this axis can be distorted during acquisition, can the authors provide some assurance of the measurements? For example, can the authors confirm that for the measurements there is representation of the apicobasal plane imaged in either the X or Y and, importantly, that they are similar/consistent with those measured on the Z-plane?

- What happens to mb upon blebbistatin treatment or in tbx5a mutants?

- 3D reconstructions would aid in the visualisation of the varying cell shapes of the OC and IC.

- Fig. S3E, F labelled as 50hpf (also in the figure legend) but is stated as 48hpf in text (pg. 9, line 1).

- Page 14, last line “normal” should read “wildtype”

Reviewer #2

(Remarks to the Author)

In this manuscript, Leerberg et al. describe the cellular and actomyosin changes that occur in zebrafish ventricular cardiomyocytes between 36-48 hpf leading to divergence of the outer and inner curvatures (OC, IC). While there is already a substantial literature on this topic, what makes this study stand out is the high level of resolution and quantification, taking advantage of a host of valuable transgenic reporter, transgenic driver, and mutant strains. After defining the OC and IC domains based on nppa and mb expression, respectively, they show that from 36-48 hpf the OC cells expand along the planar axis, increasing apical surface area, while the IC cells expand mostly along the apicobasal axis, becoming taller and rounder. This is correlated primarily with F-actin enrichment at the basal surface of OC especially proximal near the AVC, while F-actin remains distributed throughout the IC cells. OC planar expansion is shown to be MNII dependent, and disrupted by dysregulation of myl9, lack of blood flow, or mutation of tbx5a. The study makes use of blastomere transplantation to show that at least some aspects of regulating actomyosin distribution are cell autonomous. Overall, this is an excellent description for how localized cellular distribution of the actomyosin machinery drives cellular morphology of OC cells leading to organ curvature. The authors also make some interesting observations on cellular protrusions that suggest a possible link to underlying ECM, although this is quite speculative.

Overall, this is a comprehensive and highly detailed analysis of the early morphological events leading to heart looping at a critical developmental stage. It is entirely convincing and suitable for publication in Nature Communications. I have no major issues. The numbers of samples analyzed, and level of resolution is truly impressive. There are some caveats, since for imaging purposes the hearts are manipulated ex vivo, which probably should be mentioned clearly up front. It would also be helpful to indicate what eventually happens to these hearts (assuming they were not dissected but embryos left to develop). For example, presumably the DN-myl9 and CA-myl9 hearts go on to loop etc. since these appear to be established lines. Why do they recover? Likewise for the drug-treated larvae, do they recover? The authors should also check to align the text and figure panels for congruency (36, 37, 48, 50 hr?). These are very minor issues and the next most interesting mechanistic studies, such as identifying the tbx5-regulated genes that mediate these phenomena are arguably beyond the scope of this excellent paper.

Reviewer #3

(Remarks to the Author)

The study provides a comprehensive exploration of how actomyosin dynamics during cell shape changes in cardiomyocytes during ventricular chamber curvature formation. The findings that OC and IC cardiomyocytes display distinct actomyosin organization and morphologies—OC cells becoming squamous and expanding along the planar axis, while IC cells become cuboidal and elongate along the apicobasal axis—are insightful. These shape changes are preceded by specific actomyosin patterns and are influenced by both intrinsic factors (such as actomyosin and Tbx5a) and extrinsic cues (like blood flow). The methods section is thorough, and the complexity of the data adds to the study's depth. However, there are some aspects where the data remain inconclusive or unclear, particularly in the subcellular localization patterns of actomyosin due to limitations in image resolution and analysis. Since this paper addresses regional patterning, higher-quality images would improve the clarity of the findings.

A significant concern I have is the lack of mention of the extracellular matrix (ECM), particularly cardiac jelly, as an extrinsic source of curvature. While the study focuses comprehensively on the intrinsic role of actomyosin, and references other extrinsic sources like blood flow, it would be beneficial to at least mention the ECM, as it likely plays a role in curvature formation, similar to its role in gut looping or atrioventricular canal (AVC) formation during the same developmental stages.

Additionally, while actomyosin regional patterning may precede cell shape changes, these changes could still be a response to extrinsic forces (like the ECM) rather than being causal to curvature induction. This distinction should be addressed more explicitly to clarify the contribution of actomyosin dynamics.

The mechanism underlying the differences in morphology between OC and IC cells in response to changes in actomyosin activity also remains unclear. The expected roles of dominant-negative and constitutively active myosin variants, disrupting or increasing contractility, need further clarification, especially in terms of how these changes lead to the observed elongation and cuboidal phenotypes.

I recommend that the number of replicates for each experiment be stated clearly.

Specific Suggestions:

Figure 1:

- Page 8, line 6: The statement "Cells allocate their 'new volume' to different axes" raises the question of whether cell division is occurring. If it is, it would be helpful to address whether cell division rates are comparable in OC and IC cells.
- Page 9, lines 30–31; Page 10, line 1: It would be valuable to include a side-by-side comparison of actomyosin localization and cell shape divergence over time within the same embryos and cells. This would help to better connect the spatial and temporal patterns of actomyosin dynamics with the observed morphological changes.

Figure S6:

The authors suggest that actomyosin is involved in planar expansion, but traditionally, higher actomyosin levels are associated with contraction, especially when non-muscle myosin II (NMII) is involved (as mentioned in the discussion). The result showing that Bleb treatment causes excessive planar expansion could be explained by the myosin motor being locked to F-actin, behaving more like a cross-linker rather than a motor. This suggests that myosin may be playing a contractile role rather than an expansive one.

Figure 3:

The data presented here are not conclusive, and the differences between the experimental conditions seem too small, especially given that the quantification was done manually. It would be helpful to express the mutant transgenes in conditions where Rho kinase (Rok) or MLCK (or another kinase involved here) is knocked down, as the DN and CA versions of myosin may not dilute endogenous wild-type myosin sufficiently.

Figure S7:

The effect on F-actin is quite surprising. The justification for the quantitative comparison should be more robust—were the immunostaining experiments performed on samples processed in the same way? Are the transgenes in the same genetic background? It would be useful to use a Utr(ABD) line or another method to confirm the findings

Figure 4:

The authors claim that transplantation addresses the cell-intrinsic versus tissue-scale role of actomyosin; however, the transplantation leads to large groups of mutant cells rather than the salt-pepper labeling that would allow for a more precise evaluation. Using plasmid injection (rather than transplantation) may be more revealing of whether neighboring wild-type cells influence the shape of the mutant cells or not.

Figure 5:

The findings regarding the total levels of F-actin remain unclear. A more detailed explanation and justification for the quantitative comparison would be helpful.

Figure 8:

Including an illustration or phase diagram that summarizes the effects of various actomyosin and Tbx perturbations on apical-basal length and apical surface area would greatly enhance the clarity and the working model.

Version 1:

Reviewer comments:

Reviewer #1

(Remarks to the Author)

Whilst the authors have not completed all of the recommended revisions made, they have provided sound justification for why certain experiments were not, or could not, be performed. However, the revision experiments they have completed have

enhanced the work and improved the manuscript overall. I also raised concerns over some technical aspects (such as measurements in X-Y versus X-Z) and the response was thorough, clear and alleviated my concerns. In sum, they have addresses sufficient concerns that I now consider the manuscript suitable for publicaiton.

Reviewer #2

(Remarks to the Author)

The authors did a reasonable job of responding to critiques and the manuscript is improved by the addition of some data and clarification in the text. I have no further issues and can recommend publication.

Reviewer #3

(Remarks to the Author)

The authors have addressed my concerns and the manuscript has significantly improved.

Response to Reviewers' Comments – Leerberg et al., NCOMMS-25-33229

We are very grateful to all three of the reviewers for their positive feedback regarding our manuscript. We also greatly appreciate the reviewers' thoughtful suggestions for strategies to strengthen our manuscript's value. We have now modified our submission in accordance with their input, both by adding new data and by adjusting the manuscript. Our revised submission includes five new supplementary figures (Supplementary Figures 8, 10, 11, 12, 13), as well as adapted versions of previous figures (Figures 3-7 and Supplementary Figures 7 and 9). We have highlighted the new and updated content in our revised manuscript and figure legends, although we have not highlighted formatting changes (such as renumbered figures or relabeled panels). Altogether, we feel that the suggested changes have substantially enhanced the significance and clarity of our manuscript, and we thank the reviewers for their assistance with these improvements. Our point-by-point responses to the reviewers' comments are assembled below, with the reviewer comments represented verbatim in blue text and our responses provided in black text.

Reviewer #1 (Remarks to the Author):

In this study, Leerberg et al investigate the influence of the actomyosin network on cardiomyocyte cell shape during curvature morphogenesis, with specific emphasis on apicobasal organisation of the cytoskeleton. Using cardiomyocyte morphometric analysis at different times in cardiac development, the authors describe that OC and IC cells diverge in their morphology and show F-actin and p-Myosin organisation precedes cell shape changes, presumably driving ventricular wall curvature. Whilst this is perhaps unsurprising, it is novel and has (to our knowledge) not been demonstrated in this process before. Using pharmacological treatments, transgenics, and mutant analyses combined with cell transplant assays, they investigate the actomyosin network during this process and how it is regulated by both intrinsic and extrinsic factors. The experiments are elegantly performed and analyses carefully conducted, with the interpretation of the data mostly appropriate.

We are grateful to Reviewer #1 for their positive assessment of the novelty of our work and the quality of our experimental design, execution, and analysis.

There are two major concerns with the work that require addressing:

1. The work using the myl9 transgenics (DN and CA) are not convincing and at times contradictory.

There are several aspects to this:

We thank the reviewer for their series of important questions regarding our use of the *DN-myl9* and *CA-myl9* transgenes. As noted below, Reviewers #2 and #3 also asked questions about these transgenes, and we greatly appreciate the opportunity to clarify the utility of these tools for our studies.

a. The generation of stable DN and CA *myl9* transgenics is surprising. If these are major modifiers of the actomyosin network, it's surprising they are viable (for e.g., sustained blebbistatin exposure would likely be lethal). Is there a phenotype in the adult fish? Are they healthy and 100% viable? Is there a difference in the curvature of the heart? If not, it could be argued that these are not the correct model to address central questions of the manuscript.

The stable transgenic lines *Tg(myl7:DN-myl9-eGFP)^{bns333Tg}* and *Tg(myl7:CA-myl9-eGFP)^{bns332Tg}* were generated by our co-authors Drs. Priya and Stainier and have been utilized in their previous publications (Priya et al., *Nature*, 2020; Albu et al., *Nature Communications*, 2024). Although cardiac chamber morphology and function have not been carefully examined in adult transgenic fish, these prior studies have established that adult transgenics are generally healthy, viable, and able to reproduce. It is important to note that these transgenes are expressed only in the myocardium, and so they do not affect the cytoskeletal machinery in other tissues, as treatment with Blebbistatin would. Even so, we agree that it would be surprising for transgenics expressing very high levels of myocardial *DN-myl9* or *CA-myl9* expression to be viable. We therefore presume that the impact on the myocardial actomyosin network in the *Tg(myl7:DN-myl9-eGFP)^{bns333Tg}* and *Tg(myl7:CA-myl9-eGFP)^{bns332Tg}* transgenic lines is relatively modest. Indeed, our revised manuscript includes new data indicating that, while pMyosin levels are reduced in *DN-myl9* hearts, some pMyosin still remains (Supplementary Fig. 8; p. 11); perhaps the survival of *DN-myl9* fish is due to this residual non-muscle myosin II (NMII) activity. Despite the presence of residual NMII activity in *DN-myl9* transgenic hearts, we think that the effect of *DN-myl9* expression on both the levels and organization of F-actin (Supplementary Figs. 8 and 9; p. 11) position the *Tg(myl7:DN-myl9-eGFP)^{bns333Tg}* transgenic line as a reasonable and valuable tool for investigating the influence of the actomyosin cytoskeleton on ventricular cardiomyocyte morphology. (As described in the next response below, we have decided to remove experiments utilizing *Tg(myl7:CA-myl9-eGFP)^{bns332Tg}* from this manuscript.)

b. *DN-myl9* and *CA-myl9* are used to investigate the role of NMII in OC and IC cardiomyocyte morphologies. *DN-myl9* weakens the actomyosin machinery while *CA-myl9* strengthens it. Yet, the use

of both transgenes results in similar phenotypes such as decreased apical surface area and F-actin levels. Could the authors comment on this?

This is an interesting question, and its answer probably relates to the multiple cellular mechanisms at play. For example, cardiomyocyte morphologies are likely to be influenced both by the cell-intrinsic roles of the actomyosin machinery and by the cell-extrinsic forces exerted upon a cell by its neighbors. In transgenic hearts, both the intrinsic and extrinsic mechanisms are altered, and the differential interplay between these components could yield decreased apical surface area of OC cardiomyocytes in both the *DN-myl9* and *CA-myl9* contexts. Additionally, as shown in Supplementary Fig. 6, the timing of actomyosin modulation is an important factor, with NMII activity seeming to play different roles between 24-36 hpf and between 36-48 hpf. Since the *myl7* promoter drives transgene expression throughout the stages examined, it is possible that the effects of transgene expression during the later interval could mask detection of earlier effects (or vice versa) (p. 21).

Taking into account this input from Reviewer #1, together with related feedback from Reviewer #3 (discussed below), we have decided to remove all experiments utilizing the *CA-myl9* transgene from our manuscript in order to streamline its overall narrative. Although we feel that the *CA-myl9* data highlight interesting aspects of the relevant cell biology, we recognize that their interpretation is complex and that they are not needed to support our primary conclusions. In our revised manuscript (Fig. 3 and Supplementary Figs. 7-10; pp. 10-12), we focus on the use of the *DN-myl9* transgene to demonstrate that reduction of NMII activity alters F-actin organization and inhibits the planar expansion of OC cardiomyocytes.

c. What's the explanation for the reduced fluorescence of F-actin? Could it be squelching or a similar effect that is different from intended?

We appreciate Reviewer #1's curiosity about the reduced phalloidin signal observed in hearts expressing *DN-myl9*; this is echoed below in Reviewer #1's third minor comment, as well as in two comments from Reviewer #3. Although our studies have not yet elucidated the precise mechanisms responsible for the reduced levels of phalloidin signal in *DN-myl9* transgenic hearts, we do not think that this observation is an imaging artifact. All hearts were treated and imaged similarly, regardless of experimental group, and care was taken to avoid loss of fluorescent signal (e.g. avoiding unnecessarily long or intense laser exposure). Additionally, our revised manuscript includes new data that help to address this topic

(Supplementary Fig. 8; p. 11). Specifically, in addition to exhibiting reduced levels of F-actin, *DN-myf9* transgenic hearts display a coincident reduction in pMyosin levels (suggesting reduced NMII activity) and, by contrast, display levels of G-actin that are comparable to the levels in wild-type siblings (Supplementary Fig. 8). These new data suggest that, although actin monomers are readily available in *DN-myf9* transgenic hearts, they are not being polymerized into filaments to the same degree as in wild-type siblings (p. 11). This is possibly due to the reduced NMII activity in *DN-myf9* hearts, as NMII is known to affect the F-actin landscape in myriad ways (e.g. references #68-#76 in our revised manuscript).

d. What are the expression levels of the transgenics like? Are they weaker than the mosaic or injected F0? Could there be some silencing/inhibition of the transgene occurring that might account for the subtlety of the phenotypes?

In the transgenic line *Tg(myf7:DN-myf9-eGFP)^{bns333Tg}*, the DN-myf9 protein is fused to eGFP, and we find that eGFP fluorescence is easily visible in the heart by ~24 hpf and that fluorescence levels are reasonably consistent among transgenic embryos. Additionally, this level of eGFP fluorescence appears comparable to the level seen in donor-derived transgenic cardiomyocytes in mosaic hearts generated via blastomere transplantation (Fig. 3). In contrast, transgene injection generates F0 embryos with mosaic hearts that exhibit highly variable amounts of fluorescence from cell to cell (Supplementary Fig. 10); this broad range of mosaicism is common in F0 zebrafish embryos (e.g. Fisher et al., *Nature Protocols*, 2006). Therefore, we agree with Reviewer #1 that the level of *DN-myf9* expression in our transgenic embryos is lower than can sometimes be found in injected F0 embryos. As discussed above (Reviewer #1, comment #1a), we suspect that these relatively modest *DN-myf9* expression levels are likely to account for the viability of *Tg(myf7:DN-myf9-eGFP)^{bns333Tg}* transgenic embryos. Although higher levels of expression can be achieved in injected F0 embryos, we feel that the consistency of *DN-myf9* expression levels found in transgenic embryos is valuable for our studies. Even so, motivated by the input from Reviewer #1 and Reviewer #3, our revised manuscript also incorporates new data in which we complement our blastomere transplantation approach with analysis of transgene-injected F0 embryos (Supplementary Fig. 10; p. 12). Satisfyingly, we found that transgene-expressing OC cardiomyocytes in injected F0 embryos exhibited planar expansion defects similar to the defects found in the donor-derived OC cardiomyocytes in our blastomere transplantation experiments (Fig. 3 and Supplementary Fig. 10; p. 12).

e. Some effects in the mosaic analysis are opposite to that observed in the transgenic. For e.g. the circularity of stable CA *myl9* is decreased in the OC and yet increased in the mosaic analysis. Could the authors comment on this?

As alluded to above (Reviewer #1, comment #1b), differences between cell morphologies in transgenic hearts and cell morphologies in mosaic hearts could reflect the differential impacts of the cell-intrinsic and cell-extrinsic roles played by the actomyosin machinery. In a transgenic heart, all neighboring cardiomyocytes express the transgene, providing less (*DN-myl9*) or more (*CA-myl9*) NMII activity than wild-type neighbors would. Even so, we note that the effects of *DN-myl9* expression are relatively comparable in transgenic hearts (Supplementary Fig. 7) and in mosaic hearts (Fig. 3): in both scenarios, inhibition of NMII activity reduces the planar expansion of OC cardiomyocytes.

Since the combination of cell-intrinsic and cell-extrinsic influences complicates the interpretation of phenotypes seen in transgenic hearts, we have decided to emphasize the significance of our *DN-myl9* mosaic analysis (Fig. 3) in our revised manuscript, and we have moved our analysis of *DN-myl9* transgenic hearts to the Supplementary Material (Supplementary Fig. 7). Additionally, as mentioned above (Reviewer #1, comment #1b), we have decided to remove all experiments utilizing the *CA-myl9* transgene from our manuscript in order to streamline its overall narrative.

2. Whilst this is an elegant study, it is brief, which limits the contribution. Could the authors take the work a little further? For example, is there any information about what Tbx5a might be regulating to control actomyosin organisation? Are there any Formins, Rho GTPases, etc that are known to be downstream of Tbx5? Alternatively, the observation of the basal projections is intriguing and there is a lot of speculation in the Discussion about how this might be regulated. Could this be tested to some extent?

We greatly appreciate Reviewer #1's interest in these important future directions for our studies. We are certainly keen to understand the factors that act downstream of Tbx5a to control actomyosin organization, and we note that existing transcriptomic analyses do suggest that certain actomyosin-associated genes, such as *EZRIN* (Kathiriya et al., *Developmental Cell*, 2021) or *formin like 2b* (Pawlak et al., *Genome Research*, 2019), could be regulated by Tbx5. Although we are actively investigating potential downstream factors, thorough analysis of each candidate will require a significant amount of

additional experimentation, beyond the scope and timeframe for revision of our current manuscript. We therefore respectfully decline to include this research direction within our revised submission. Similarly, we appreciate Reviewer #1's curiosity about the regulation of the basal projections shown in Supplementary Figures 16 and 17. We are actively investigating this open question, and we believe that it may ultimately relate to the questions raised by Reviewer #3 regarding ECM involvement (see below). However, the mechanisms underlying these intriguing cell behaviors are likely to be complex, and their elucidation will require extensive experimentation that lies beyond the scope of this revision. Again, we respectfully decline to include this research direction in our revised manuscript.

Minor comments:

- The authors show that there are cell shape differences between the OC and the IC. This appears to be driven by the organisation of the actomyosin network. While manipulation of the actomyosin network (by DN-myf9, CA-myf9, blood flow, *tbx5a* mutants etc.) results in reduced planar spread in OC cells, localisation of F-actin and in some cases pMyo in the proximal OC is the most affected. Looking more closely in WT hearts, the expression pattern of F-actin and pMyo (fig S3 and figS4D) in the distal OC appears more similar to the IC. This begs the question – are there cell shape differences between the proximal and distal cells of the OC? I.e. is the proximal OC more squamous than the distal OC? Additionally, in treated samples are proximal OC cell shape more disrupted?

We thank Reviewer #1 for asking about this interesting topic. We have not detected any significant and consistent morphometric differences between wild-type OC cardiomyocytes in the regions that we define as proximal and distal at 48 hpf, and we mention this in our revised Discussion (p. 22). However, it is important to note that the cardiomyocytes found in the regions that we define as proximal and distal when analyzing cell morphologies at 48 hpf may not precisely correspond to the cardiomyocytes found in the regions that we define as proximal and distal when analyzing actomyosin organization at 36 hpf, which may mask our detection of meaningful morphological differences at 48 hpf. In future studies beyond the scope of the current manuscript, we plan to utilize timelapse imaging to track these populations of cells between 36 and 48 hpf.

Although we have not found significant differences in the morphologies of proximal and distal OC cardiomyocytes in wild-type hearts, we have also analyzed whether proximal and distal OC cells are differentially impacted in different genetic backgrounds. We did not observe significant morphometric differences between proximal and distal OC cells in *DN-myf9* transgenic or *tbx5a* mutant hearts (data

not shown). However, in *myh6* mutant hearts, proximal OC cardiomyocytes did seem to be more severely affected than distal OC cardiomyocytes (Reviewer Fig. 1). While apical surface area is affected similarly in proximal and distal regions (Reviewer Fig. 1a; Cohen's $d = 0.61$ (proximal) vs 0.72 (distal), $\Delta 0.11$), apicobasal length is more affected in the proximal region (Reviewer Fig. 1b; Cohen's $d = -0.55$ (proximal) vs -0.27 (distal), $\Delta 0.28$), as is apical circularity (Reviewer Fig. 1c; Cohen's $d = 0.53$ (proximal) vs -0.05 (distal), $\Delta 0.58$). This analysis suggests that the proximal and distal regions of the OC might be differentially sensitive to the biomechanical cues associated with blood flow, perhaps due to underlying differences in the actomyosin landscape. While we are happy to share this information with the reviewers, we prefer not to incorporate it into our revised manuscript at this time, since we feel that substantially more work will be required to understand why we only see these differences in the *myh6* mutant context (and not in the *tbx5a* mutant or *DN-myl9* transgenic contexts) and to understand why the differential effects are found for apicobasal length and circularity (and not for apical surface area).

Reviewer Figure 1. Blood flow affects cardiomyocytes in the proximal and distal regions of the OC differently.

(a-c) OC data from Fig. 4m-o, split into proximal and distal regions. For all violin plots, each dot represents an individual cell, and each black bar represents the mean. * denotes $p < 0.05$ and *** denotes $p < 0.001$, Wilcoxon test. Cohen's d values represent the effect size. Wild-type proximal OC (N=5 embryos, 142 cells); *myh6* proximal OC (N=6 embryos, n=147 cells); wild-type distal OC (N=5 embryos, 125 cells); *myh6* distal OC (N=6 embryos, n=128 cells).

- The authors posit that changes in cell shape accompany curvature formation. In several assays conducted by the authors cell shape especially in the OC is affected but its effect on characteristic chamber curvatures is not measured. Could this be addressed?

We agree with Reviewer #1 that it would be desirable to evaluate how individual cardiomyocyte morphologies contribute to the overall geometry of the ventricle, and we plan to address this in future studies. This work will involve the creation of new two-dimensional and three-dimensional metrics for chamber curvatures, establishment of the variation found in wild-type curvatures at stages between 24 and 48 hpf, and mathematical modeling of the contributions of cell shapes to curvature dimensions; therefore, this combination of pursuits lies beyond the scope of our current manuscript.

- Disruption of NMII, blood flow and *tbx5a* function result in reduced F-actin levels. Is this due to a reduction of actin expression or is actin polymerisation affected? An IF for G-actin or a western blot for actin may aid in answering this.

We thank Reviewer #1 (and Reviewer #3, below) for encouraging us to investigate this topic further. We appreciated and pursued the suggestion to examine G-actin levels, and this has led to the inclusion of three new figures in our revised manuscript (Supplementary Figs. 8, 12, and 13). As discussed above (Reviewer #1, major comment #1c), we found that the reduced levels of F-actin in *DN-myl9* transgenic hearts are accompanied by levels of G-actin that are comparable to the levels in wild-type siblings (Supplementary Fig. 8). These data suggest that, although actin monomers are readily available in *DN-myl9* transgenic hearts, they are not being polymerized into filaments to the same degree as in wild-type siblings (p. 11). Similarly, we have found that *myh6* mutants and *tbx5a* mutants exhibit reduced levels of F-actin, together with G-actin levels that are similar to wild-type (Supplementary Figs. 12 and 13). As in *DN-myl9* transgenics, these data suggest that availability of actin monomers is normal in *myh6* mutants and in *tbx5a* mutants, but that actin modifiers are not leading to normal polymerization of actin filaments in these hearts (pp. 13, 14, 20).

- Apicobasal length is measured using confocal images. In some instances, it appears as though the images have been acquired with in the X-Y plane. At other times, apicobasal axis appears to be acquired in the Z-axis. Given that this axis can be distorted during acquisition, can the authors provide some assurance of the measurements? For example, can the authors confirm that for the measurements there is representation of the apicobasal plane imaged in either the X or Y and, importantly, that they are similar/consistent with those measured on the Z-plane?

We thank Reviewer #1 for raising this important point. Our standard practice is to mount hearts laterally, with one side of the ventricle flat against the coverslip, and we have clarified this in our revised Methods

section (p. 26). In this orientation (Reviewer Fig. 2a,b), the X-Y axes of most of the OC and IC cells are perpendicular to the objective. This results in more resolvable signal in the section views used to measure apicobasal length (Reviewer Fig. 2b), compared to when hearts are mounted with the OC closest to the objective (Reviewer Fig. 2c,d). Additionally, mounting in our standard orientation allows us to visualize both the OC and IC with similar clarity and intensity (compare the ICs in Reviewer Fig. 2b and 2d). However, it is important to note that the cells in the curvatures are not all in the same plane, as the ventricle is a bulging, three-dimensional structure. Therefore, we performed an experiment in which the same three hearts were imaged in two perpendicular orientations, and we measured the apicobasal lengths of the OC cells along the midline in each set of images (Reviewer Fig. 2e). Reassuringly, we found similar apicobasal length values when comparing these perpendicular orientations (Reviewer Fig. 2e).

Reviewer Figure 2. Hearts imaged in perpendicular orientations reveal similar apicobasal lengths of OC cardiomyocytes.

(a,b) 3D reconstruction of the OC (a) and section through the ventricle (b) of a heart mounted flat against the coverslip, resulting in one side of the ventricle being closest to the objective. (c,d) 3D reconstruction of the OC (c) and section through the ventricle (d) of the same heart shown in (a) and (b), but mounted with the OC against the coverslip and closest to the objective. Sections in (b) and (d) are taken at the positions represented by the dotted red lines in (a) and (c). Scale bars = 20 μm. (e) Three hearts were imaged in two orientations each. For each of the three hearts, apicobasal (AB) length was measured (as in Fig. 1) for OC cardiomyocytes that intersected with the midline (represented by the

dotted red line in (a) and (c)). In the violin plots, each heart is represented by a different shape; and each cell measured is represented by a different color. $p > 0.05$, paired Wilcoxon test.

- What happens to mb upon blebbistatin treatment or in *tbx5a* mutants?

We have not noted any differences in the pattern of *mb* (*myoglobin*) expression in the *tbx5a* mutant ventricle at 36 hpf, suggesting that the initial patterning of IC identity may not depend upon *tbx5a* function. We have not examined *mb* expression in blebbistatin-treated embryos, and we chose not to prioritize this experiment as part of our revision efforts for this manuscript.

- 3D reconstructions would aid in the visualisation of the varying cell shapes of the OC and IC.

We appreciate Reviewer #1's interest in viewing 3D reconstructions of the morphologies of individual OC and IC cardiomyocytes. However, the methods used in the relevant experiments (Figs. 1, 4, 6) do not permit this type of visualization: these studies used *Cdh2* to label the lateral membranes of cardiomyocytes but did not include a transgene like *Tg(my17:eGFP-Hsa.HRAS)* that outlines all of the cardiomyocyte surfaces. We have therefore respectfully declined to include 3D cardiomyocyte morphology reconstructions in our revised manuscript.

- Fig. S3E, F labelled as 50hpf (also in the figure legend) but is stated as 48hpf in text (pg. 9, line 1).

We have made this correction in our revised manuscript (p. 8). Please note that the panel labeling for Supplementary Fig. 3 has changed, but we have changed "48 hpf" to "50 hpf" as needed.

- Page 14, last line "normal" should read "wildtype"

We have made this correction in our revised manuscript (p. 14).

Reviewer #2 (Remarks to the Author):

In this manuscript, Leerberg et al. describe the cellular and actomyosin changes that occur in zebrafish ventricular cardiomyocytes between 36-48 hpf leading to divergence of the outer and inner curvatures (OC, IC). While there is already a substantial literature on this topic, what makes this study stand out is

the high level of resolution and quantification, taking advantage of a host of valuable transgenic reporter, transgenic driver, and mutant strains. After defining the OC and IC domains based on *nppa* and *mb* expression, respectively, they show that from 36-48 hpf the OC cells expand along the planar axis, increasing apical surface area, while the IC cells expand mostly along the apicobasal axis, becoming taller and rounder. This is correlated primarily with F-actin enrichment at the basal surface of OC especially proximal near the AVC, while F-actin remains distributed throughout the IC cells. OC planar expansion is shown to be MNII dependent, and disrupted by dysregulation of *myl9*, lack of blood flow, or mutation of *tbx5a*. The study makes use of blastomere transplantation to show that at least some aspects of regulating actomyosin distribution are cell autonomous. Overall, this is an excellent description for how localized cellular distribution of the actomyosin machinery drives cellular morphology of OC cells leading to organ curvature. The authors also make some interesting observations on cellular protrusions that suggest a possible link to underlying ECM, although this is quite speculative.

Overall, this is a comprehensive and highly detailed analysis of the early morphological events leading to heart looping at a critical developmental stage. It is entirely convincing and suitable for publication in Nature Communications. I have no major issues. The numbers of samples analyzed, and level of resolution is truly impressive. There are some caveats, since for imaging purposes the hearts are manipulated *ex vivo*, which probably should be mentioned clearly up front. It would also be helpful to indicate what eventually happens to these hearts (assuming they were not dissected but embryos left to develop). For example, presumably the DN-*myl9* and CA-*myl9* hearts go on to loop etc. since these appear to be established lines. Why do they recover? Likewise for the drug-treated larvae, do they recover? The authors should also check to align the text and figure panels for congruency (36, 37, 48, 50 hr?). These are very minor issues and the next most interesting mechanistic studies, such as identifying the *tbx5*-regulated genes that mediate these phenomena are arguably beyond the scope of this excellent paper.

We are grateful to Reviewer #2 for their strong support of the quality, rigor, and significance of our studies. We also appreciate the minor issues raised by Reviewer #2, and we can address each of these, as follows:

- We have clarified that hearts are dissected from embryos before imaging, both in our Methods section (p. 26) and in the Fig. 1 legend.

- Above (Reviewer #1, comment #1a), we have discussed that *DN-myf9* transgenic hearts likely recover due to their remaining pMyosin (Supplemental Fig. 8).
- We have not performed washout experiments to examine how well Latrunculin B-treated or Blebbistatin-treated embryos recover. However, since these drugs affect actin or myosin dynamics throughout the embryo, it is likely that their recovery could be hindered by defects in multiple tissues.
- We have corrected issues in the congruency between our text and figure panels.

Reviewer #3 (Remarks to the Author):

The study provides a comprehensive exploration of how actomyosin dynamics during cell shape changes in cardiomyocytes during ventricular chamber curvature formation. The findings that OC and IC cardiomyocytes display distinct actomyosin organization and morphologies—OC cells becoming squamous and expanding along the planar axis, while IC cells become cuboidal and elongate along the apicobasal axis—are insightful. These shape changes are preceded by specific actomyosin patterns and are influenced by both intrinsic factors (such as actomyosin and *Tbx5a*) and extrinsic cues (like blood flow). The methods section is thorough, and the complexity of the data adds to the study's depth. However, there are some aspects where the data remain inconclusive or unclear, particularly in the subcellular localization patterns of actomyosin due to limitations in image resolution and analysis. Since this paper addresses regional patterning, higher-quality images would improve the clarity of the findings.

We are grateful to Reviewer #3 for their positive evaluation of the “comprehensive”, “insightful”, and “thorough” nature of our studies, and we also appreciate their suggestions for ways to improve the clarity and impact of our data. Regarding Reviewer #3’s request for higher-quality images, we have adjusted the sizing of several figures in our revised manuscript (Figs. 1, 2, 3, 4, 6) in an effort to improve the visibility of their details.

A significant concern I have is the lack of mention of the extracellular matrix (ECM), particularly cardiac jelly, as an extrinsic source of curvature. While the study focuses comprehensively on the intrinsic role of actomyosin, and references other extrinsic sources like blood flow, it would be beneficial to at least mention the ECM, as it likely plays a role in curvature formation, similar to its role in gut looping or atrioventricular canal (AVC) formation during the same developmental stages.

We appreciate Reviewer #3's suggestion, and we agree that the ECM likely plays an important role during curvature formation and the underlying cellular and subcellular mechanisms. In our revised manuscript, we have added text to the Discussion to highlight this likelihood (pp. 18, 20, 22).

Additionally, while actomyosin regional patterning may precede cell shape changes, these changes could still be a response to extrinsic forces (like the ECM) rather than being causal to curvature induction. This distinction should be addressed more explicitly to clarify the contribution of actomyosin dynamics.

We agree with Reviewer #3 that extrinsic forces, including the ECM, influence the cytoskeleton, and it is certainly possible that the observed differences in cytoskeletal organization, although correlated with shape change, are not causative. In our revised manuscript, we have added text to the Discussion to address these possibilities and the future work that will be needed to test them (pp. 18, 20, 22).

The mechanism underlying the differences in morphology between OC and IC cells in response to changes in actomyosin activity also remains unclear. The expected roles of dominant-negative and constitutively active myosin variants, disrupting or increasing contractility, need further clarification, especially in terms of how these changes lead to the observed elongation and cuboidal phenotypes.

Please see responses above to Reviewer #1, comments #1a, #1b, and #1c. Notably, our revised manuscript includes new data demonstrating that pMyosin levels are reduced in *DN-myl9* hearts, (Supplemental Fig. 8). Also, as discussed above, we have decided to remove all experiments utilizing the *CA-myl9* transgene from our revised manuscript.

I recommend that the number of replicates for each experiment be stated clearly.

We thank Reviewer #3 for this suggestion. In our revised manuscript, all relevant figure legends now include the number of experimental replicates, along with the number of embryos analyzed and, where appropriate, the number of cells analyzed.

Specific Suggestions:

Figure 1:

- Page 8, line 6: The statement "Cells allocate their 'new volume' to different axes" raises the question of

whether cell division is occurring. If it is, it would be helpful to address whether cell division rates are comparable in OC and IC cells.

We did not evaluate cell division rates during this set of studies. However, prior work has shown that there is very little proliferation in the zebrafish myocardium between 24-48 hpf (de Pater et al., 2009). Specifically, this study found only <2 phospho-His⁺ cardiomyocytes per heart at 30, 36, and 48 hpf, and BrdU treatment between 24-48 hpf only yielded ~16 BrdU⁺ cardiomyocytes per heart, with more than half of these located near the cardiac poles. Therefore, it is unlikely that cell division has much of an impact on OC and IC cells during the ~36-48 hpf timeframe that is the focus of our manuscript.

- Page 9, lines 30–31; Page 10, line 1: It would be valuable to include a side-by-side comparison of actomyosin localization and cell shape divergence over time within the same embryos and cells. This would help to better connect the spatial and temporal patterns of actomyosin dynamics with the observed morphological changes.

We thank Reviewer #3 for this suggestion, and we certainly appreciate the value of being able to follow actomyosin organization and cell shape changes simultaneously in the same cells over time. Unfortunately, our current imaging tools and techniques cannot accommodate the live imaging experiments necessary for these side-by-side comparisons, as the ventricle is located deep under the head during the critical stages of actomyosin dynamics studied here. However, in future studies, beyond the scope of our current manuscript, we hope to pursue this direction by adopting newly emerging techniques, such as Dr. Priya's "body explant" strategy (Jie et al., *bioRxiv*, 2025).

Figure S6:

The authors suggest that actomyosin is involved in planar expansion, but traditionally, higher actomyosin levels are associated with contraction, especially when non-muscle myosin II (NMII) is involved (as mentioned in the discussion). The result showing that Bleb treatment causes excessive planar expansion could be explained by the myosin motor being locked to F-actin, behaving more like a cross-linker rather than a motor. This suggests that myosin may be playing a contractile role rather than an expansive one.

Reviewer #3 brings up an interesting point regarding the different potential roles of NMII. We think that our data support multiple roles for NMII (pp. 10, 21): an earlier role (24-36 hpf) when NMII is involved

in constructing the F-actin landscape and actualizing the subsequent divergence of OC and IC cell shapes, and a later role (36-48 hpf) when NMII helps maintain cell shape and size via its contractile properties (as Reviewer #3 suggests). This idea is supported by the stage-dependent nature of our results with Blebbistatin: treatment from 24-36 hpf resulted in smaller cells at 48 hpf, but treatment from 36-48 hpf resulted in larger cells at 48 hpf (p. 10). It will be important in future work to be able to temporally control cytoskeletal manipulations to better understand these complexities (p. 21).

Figure 3:

The data presented here are not conclusive, and the differences between the experimental conditions seem too small, especially given that the quantification was done manually. It would be helpful to express the mutant transgenes in conditions where Rho kinase (Rok) or MLCK (or another kinase involved here) is knocked down, as the DN and CA versions of myosin may not dilute endogenous wild-type myosin sufficiently.

We appreciate Reviewer #3's interest in this topic, which is shared with Reviewer #1. We agree that the impact of the *DN-myf9* transgene on NMII activity is an important question. As discussed above in response to Reviewer #1, comment #1a, we have addressed this by examining the levels of pMyosin in *DN-myf9* transgenic hearts, where we found a significant reduction in comparison to wild-type hearts (Supplementary Fig. 8; p. 11). We reason that this reduction, coupled with the observed effects of the *DN-myf9* transgene on total F-actin and on F-actin organization (Supplementary Figs. 8 and 9; p. 11), positions the *DN-myf9* transgene as a valuable tool for investigating the influence of the actomyosin cytoskeleton on ventricular cardiomyocyte morphology. Additionally, in our revised submission, we have decided to emphasize the significance of our *DN-myf9* mosaic analysis (now Fig. 3 of our revised manuscript), and we have moved our analysis of *DN-myf9* transgenic hearts to the Supplementary Material (Supplementary Fig. 7). Finally, as mentioned above (Reviewer #1, comment #1b), we have decided to remove all experiments utilizing the *CA-myf9* transgene from our manuscript in order to streamline its overall narrative.

Figure S7:

The effect on F-actin is quite surprising. The justification for the quantitative comparison should be more robust—were the immunostaining experiments performed on samples processed in the same way? Are the transgenes in the same genetic background? It would be useful to use a Utr(ABD) line or

another method to confirm the findings

We appreciate Reviewer #3's interest in this point, which is again shared with Reviewer #1 (comment #1c above). In our revised manuscript, we address the F-actin levels in *DN-myl9* transgenics in Supplementary Fig. 8. The samples examined in Supplementary Fig. 8 were indeed processed in a parallel fashion (see Methods description on pp. 23-25), and the *DN-myl9* transgenic embryos and their wild-type siblings were in the same genetic background. As described above (Reviewer #1, comment #1c), we found that in addition to exhibiting reduced levels of F-actin, *DN-myl9* transgenic hearts display a coincident reduction in pMyosin levels (suggesting reduced NMII activity) and, by contrast, display levels of G-actin that are comparable to the levels in wild-type siblings (Supplementary Fig. 8). These new data suggest that, although actin monomers are readily available in *DN-myl9* transgenic hearts, they are not being polymerized into filaments to the same degree as in wild-type siblings (p. 11), possibly due to the reduced NMII activity in *DN-myl9* hearts. Although we did not have a Utr(ABD) line available for these studies, we hope that our G-actin staining approach will help to address the reviewer's queries.

Figure 4:

The authors claim that transplantation addresses the cell-intrinsic versus tissue-scale role of actomyosin; however, the transplantation leads to large groups of mutant cells rather than the salt-pepper labeling that would allow for a more precise evaluation. Using plasmid injection (rather than transplantation) may be more revealing of whether neighboring wild-type cells influence the shape of the mutant cells or not.

We appreciate Reviewer #3's astute observation regarding the potential influence of neighboring cells on the shape of donor-derived, transgene-expressing cells. However, in our experiments, we were pleased to be able to achieve a wide variety of clone sizes (ranging from 1 donor-derived cell to 33 donor-derived cells), and not just large groups of donor-derived cells (Reviewer Fig. 3). Furthermore, we found no correlation between clone size and any of the morphometrics tested (For OC cells: apical surface area: $R^2 = 0.02$, $p = 0.32$; apicobasal length: $R^2 = 0.01$, $p = 0.36$; apical circularity: $R^2 = 0.04$, $p = 0.12$. For IC cells: apical surface area: $R^2 < 0.01$, $p = 0.91$; apicobasal length: $R^2 = 0.03$, $p = 0.39$; apical circularity: $R^2 = 0.04$, $p = 0.3$). Even so, we have performed the analysis of transgene-injected F0 embryos, as suggested by Reviewer #3 (and Reviewer #1), and we have incorporated these data into our revised manuscript (Supplementary Fig. 10; p. 12). Satisfyingly, we have found that transgene-

expressing OC cardiomyocytes in injected F0 embryos exhibited planar expansion defects similar to the defects found in the donor-derived OC cardiomyocytes in our blastomere transplantation experiments (Fig. 3 and Supplementary Fig. 10; p. 12). As mentioned above (Reviewer #1, comment #1d), we noted substantial variation in the expression levels of transgene expression in our injected embryos, which is common in F0 zebrafish embryos (e.g. Fisher et al., *Nature Protocols*, 2006). Because we did not observe this variation in the mosaic hearts generated by transplantation, we generally feel that the transplantation approach is a more controlled strategy for mosaic analysis.

Reviewer Figure 3. Clone sizes from *Tg(myl7:DN-myl9-eGFP)* into wild-type transplantation experiments.

Violin plots are as in Fig. 3j. Individual datapoints for wild-type host-derived cells have been removed; individual datapoints for *DN-myl9* donor-derived cells are colored and sized based on clone size. Note that cells from different clone sizes, ranging from 1 donor-derived cell to 33 donor-derived cells, are distributed broadly along the y-axis.

Figure 5:

The findings regarding the total levels of F-actin remain unclear. A more detailed explanation and justification for the quantitative comparison would be helpful.

We are unsure of exactly which elements of our original Figure 5 are being addressed by Reviewer #3 in this comment. However, we suspect that Reviewer #3 may share Reviewer #1's curiosity about the reduced F-actin levels observed in *myh6* mutant and *tbx5a* mutant hearts. As described above (Reviewer #1, minor comment #3), we have incorporated the examination of G-actin levels into our revised

manuscript (Supplementary Figs. 12 and 13). Specifically, we have found that *myh6* mutants and *tbx5a* mutants exhibit reduced levels of F-actin, together with G-actin levels that are similar to wild-type (Supplementary Figs. 12 and 13). As in *DN-myl9* transgenics, these data suggest that availability of actin monomers is normal in *myh6* mutants and in *tbx5a* mutants, but that actin modifiers are not properly executing the normal polymerization of actin filaments in these hearts (pp. 13, 14, 20).

Figure 8:

Including an illustration or phase diagram that summarizes the effects of various actomyosin and Tbx perturbations on apical-basal length and apical surface area would greatly enhance the clarity and the working model.

We thank Reviewer #3 for the suggestion to include an additional schematic in our revised manuscript. We have added a new figure (Supplementary Fig. 11; pp. 13, 14, 21, 22, 31, 32) that compares the mean apical surface areas and mean apicobasal lengths of OC and IC cardiomyocytes at different developmental stages and with different genotypes. Altogether, it nicely illustrates the divergence of wild-type OC and IC cell morphologies between 37 and 48 hpf, as well as the differences in OC cell morphologies found in *myh6* mutants and *tbx5a* mutants at 48 hpf. While we are unsure whether this schematic incorporates all of the information that Reviewer #3 had in mind, we feel that it will be useful for the readers of our revised manuscript.